# WARM DIFFUSION: RECIPE FOR BLUR-NOISE MIXTURE DIFFUSION MODELS

**Hao-Chien Hsueh    Wen-Hsiao Peng    Ching-Chun Huang**
National Yang Ming Chiao Tung University, Taiwan

## ABSTRACT

Diffusion probabilistic models have achieved remarkable success in generative tasks across diverse data types. While recent studies have explored alternative degradation processes beyond Gaussian noise, this paper bridges two key diffusion paradigms: hot diffusion, which relies entirely on noise, and cold diffusion, which uses only blurring without noise. We argue that hot diffusion fails to exploit the strong correlation between high-frequency image detail and low-frequency structures, leading to random behaviors in the early steps of generation. Conversely, while cold diffusion leverages image correlations for prediction, it neglects the role of noise (randomness) in shaping the data manifold, resulting in out-of-manifold issues and partially explaining its performance drop. To integrate both strengths, we propose Warm Diffusion, a unified Blur-Noise Mixture Diffusion Model (BNMD), to control blurring and noise jointly. Our divide-and-conquer strategy exploits the spectral dependency in images, simplifying score model estimation by disentangling the denoising and deblurring processes. We further analyze the Blur-to-Noise Ratio (BNR) using spectral analysis to investigate the trade-off between model learning dynamics and changes in the data manifold. Extensive experiments across benchmarks validate the effectiveness of our approach for image generation.

## 1 INTRODUCTION

Diffusion probabilistic models (Sohl-Dickstein et al., 2015; Ho et al., 2020; Nichol & Dhariwal, 2021; Song et al., 2022) have gained significant attention for their ability to learn data distributions through denoising, leading to impressive generation quality. These generative models typically employ a stochastic process that gradually transforms complex data distributions into simpler forms by adding a small amount of Gaussian noise in each forward iteration, eventually arriving at a simple Gaussian distribution. The reverse process involves using a neural network to model the score (Hyvärinen & Dayan, 2005) of a noise-level-dependent marginal distribution, iteratively adapting the denoised samples to recover the input data distribution. However, the process of learning this score estimator is domain-agnostic, focusing solely on recovering the underlying signal by removing Gaussian noise without considering the inherent properties of the modeled data. While this universal approach is effective for various data modalities, we argue that it leaves room for improvement in modeling images. Specifically, it overlooks the strong correlation between high-frequency image detail and low-frequency structures—a relationship we term spectral dependency. This correlation suggests that an efficient image generation process should progress from common low-frequency components to diverse high-frequency detail.

Recently, a large number of studies (Bansal et al., 2022; Daras et al., 2022; Rissanen et al., 2023; Hoogeboom & Salimans, 2024; Luo et al., 2023; Delbracio & Milanfar, 2024; Liu et al., 2023; Yue et al., 2023; Liu et al., 2024) explored various alternatives to the conventional noise-driven forward/reverse process of diffusion models, with the aim of accelerating the reverse process and solving specific inverse or image translation problems. Some of these techniques adopt a cold diffusion process, which replaces the stochastic Gaussian degradation process with deterministic image transformations, e.g. blurring. These advancements underscore the evolving nature of diffusion probabilistic models in formulating the forward/reverse process. Although these methods work well in solving specific restoration tasks, most of them struggle with generating diverse and high-quality samples, as compared to typical noise-driven hot diffusion models.

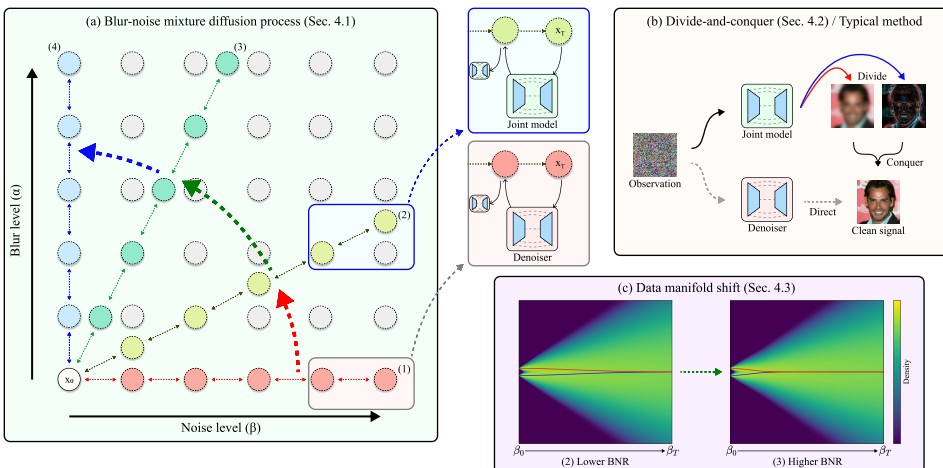

Figure 1: Illustration of Warm Diffusion, the proposed two-pronged diffusion process. (a) Graphical models of the proposed blur-noise mixture diffusion processes, offering flexibility in selecting blur and noise levels, thereby enabling a smooth transition between (1) Hot Diffusion and (4) Cold Diffusion. (b) The proposed divide-and-conquer strategy employs a joint model for denoising and deblurring. It recovers the blurry signal obscured by noise and restores missing high-frequency detail while explicitly accounting for the spectral dependency of images. (c) Data manifolds for the two diffusion processes at varying blur-to-noise ratios (BNR). The red and blue lines represent the means of Gaussian distributions derived from samples sharing low-frequency content but differing in high-frequency detail. As BNR increases, the data manifolds shift and merge at earlier stages, as blurring filters out high-frequency detail, leaving only the shared low-frequency signal. This shift may lead to out-of-manifold issues, discussed in detail in Sec. 4.3.

As shown in Fig. 1, we revisit the design of diffusion probabilistic models, expanding the pure denoising process to a joint denoising and deblurring approach. We discuss the limitations of existing hot and cold diffusion methods, particularly in terms of network learning and data manifold modeling. By taking into account the spectral dependency of images and the diversity introduced by Gaussian noise, we propose a method that balances blurring and noise levels. This enhances the learning process of the decoding neural network, maintains the diversity of the data manifold, and ultimately improves the quality of generated images. Our approach, targeting image generation, has the following contributions:

- We propose a warm diffusion process that combines blurring and noise in the forward process. The scheme allows flexible control over blur and noise levels, enabling joint deblurring and denoising to enhance image generation quality.

- We introduce the new concept of Blur-to-Noise Ratio (BNR) control and show that increasing the BNR (leaning toward cold diffusion) simplifies model learning by leveraging spectral dependency effectively. However, this also increases the risk of samples deviating from the data manifold during the reverse process.

- We select the BNR by analyzing the spectra of images and Gaussian white noise. The difference in their power spectral densities guides us to find a suitable BNR that balances two key factors: preserving the integrity of the data manifold and simplifying neural network learning.

- Extensive experiments across various datasets show that our approach outperforms state-of-the-art diffusion methods in terms of image generation quality.

## 2 RELATED WORKS

### 2.1 DIFFUSION PROBABILISTIC MODELS FOR IMAGE GENERATION

Diffusion models (Ho et al., 2020) consist of two key processes: a forward process that progressively transforms the data distribution into a Gaussian distribution by adding noise and a reverse process

that learns to denoise and recover the original data distribution. Various noise scheduling strategies have been explored across different studies. IDDPM (Nichol & Dhariwal, 2021) introduced a cosine noise schedule that gradually destroys the signal at a slower rate. Score-SDE (Song et al., 2021) provided a unified framework that represents models like SMLD (Song & Ermon, 2019) and DDPM (Ho et al., 2020) within a continuous state space, using different discretizations and distinct Stochastic Differential Equations (SDEs). EDM (Karras et al., 2022) further elucidates the design space of diffusion models. Extensive discussions within this work provide detailed analyses of training and sampling strategies, including preconditioning the network's input and output and adjusting the loss function. These enhancements, which consider neural network properties by maintaining unit variance for the model's input and output and mitigating large gradient variations, have led to significantly improved results. However, these improvements primarily address challenges posed by Gaussian noise without leveraging key image properties, such as spectral dependency, and they overlook the potential of alternative corruption processes.

## 2.2 DIFFUSION-LIKE MODELS WITH VARIATIONS OF THE CORRUPTION PROCESS

Recently, researchers have explored modifications to the corruption process within diffusion models. Cold Diffusion (Bansal et al., 2022) replaced traditional noise-based transition functions with transformations such as blur, masking, and pixelation. This innovation created a diffusion-like generative model by inverting arbitrary image transforms; however, results indicate that cold diffusion struggles to maintain high-quality outcomes. Concurrently, IHDM (Rissanen et al., 2023) introduced a progressive blurring process, where the model learns to iteratively restore blurred signals, essentially acting as the "inverse" of heat dissipation. Blurring Diffusion (Hoogeboom & Salimans, 2024) established a connection between IHDM and Gaussian diffusion (Ho et al., 2020), demonstrating that IHDM can be interpreted as a form of Gaussian diffusion in the frequency domain, albeit with different schedules across frequency bands. By integrating the blur and noise schedules from both IHDM and iDDPM (Nichol & Dhariwal, 2021), Blurring Diffusion achieved better generation quality compared to IHDM, though at the cost of more training iterations. However, these aforementioned approaches primarily focus on the mean transition function and lack a global perspective that jointly considers the role of Gaussian noise within the diffusion process, which contributes to a drop in performance.

## 3 PRELIMINARY: DENOISING DIFFUSION IMPLICIT MODELS

Considering a class of inference distributions, indexed by vector $\sigma$:

$$q_\sigma(\boldsymbol{x}_{1:T}|\boldsymbol{x}_0) \coloneqq q_\sigma(\boldsymbol{x}_T|\boldsymbol{x}_0) \prod_{t=2}^{T} q_\sigma(\boldsymbol{x}_{t-1}|\boldsymbol{x}_t, \boldsymbol{x}_0), \tag{1}$$

where given $q_\sigma(\boldsymbol{x}_T|\boldsymbol{x}_0) = \mathcal{N}(\sqrt{\alpha_T}\boldsymbol{x}_0, (1-\alpha_T)\boldsymbol{I})$, there exists a family of posterior distributions for all $t > 1$:

$$q_\sigma(\boldsymbol{x}_{t-1}|\boldsymbol{x}_t, \boldsymbol{x}_0) = \mathcal{N}(\sqrt{\alpha_{t-1}}\boldsymbol{x}_0 + \sqrt{1-\alpha_{t-1}-\sigma_t^2} \cdot \frac{\boldsymbol{x}_t - \sqrt{\alpha_t}\boldsymbol{x}_0}{\sqrt{1-\alpha_t}}, \sigma_t^2\boldsymbol{I}), \tag{2}$$

such that $q_\sigma(\boldsymbol{x}_t|\boldsymbol{x}_0) = \mathcal{N}(\sqrt{\alpha_t}\boldsymbol{x}_0, (1-\alpha_t)\boldsymbol{I})$ for all $t$, which matches the marginals as DDPM (Ho et al., 2020).

The trainable generative process is defined as Markovian where each $p_\theta(\boldsymbol{x}_{t-1}|\boldsymbol{x}_t)$ aims to utilized the knowledge of $q_\sigma(\boldsymbol{x}_{t-1}|\boldsymbol{x}_t, \boldsymbol{x}_0)$. In a sense, given a noisy observation $\boldsymbol{x}_t$, we first predict a denoised observation $D_\theta^{(t)}(\boldsymbol{x}_t)$, and then obtain $\boldsymbol{x}_{t-1}$ with $q_\sigma(\boldsymbol{x}_{t-1}|\boldsymbol{x}_t, D_\theta^{(t)}(\boldsymbol{x}_t))$. The learning objective can be therefore parameterized as:

$$L(D_\theta^t) = \lambda(t)\|D_\theta^t(\boldsymbol{x}_t) - \boldsymbol{x}_0\|_2^2. \tag{3}$$

## 4 BLUR-NOISE MIXTURE DIFFUSION MODELS (BNMD)

To enhance image generation, we introduce a blur-noise mixture diffusion model (BNMD). BNMD extends the state space of the diffusion process to two dimensions, controlled by the corruption

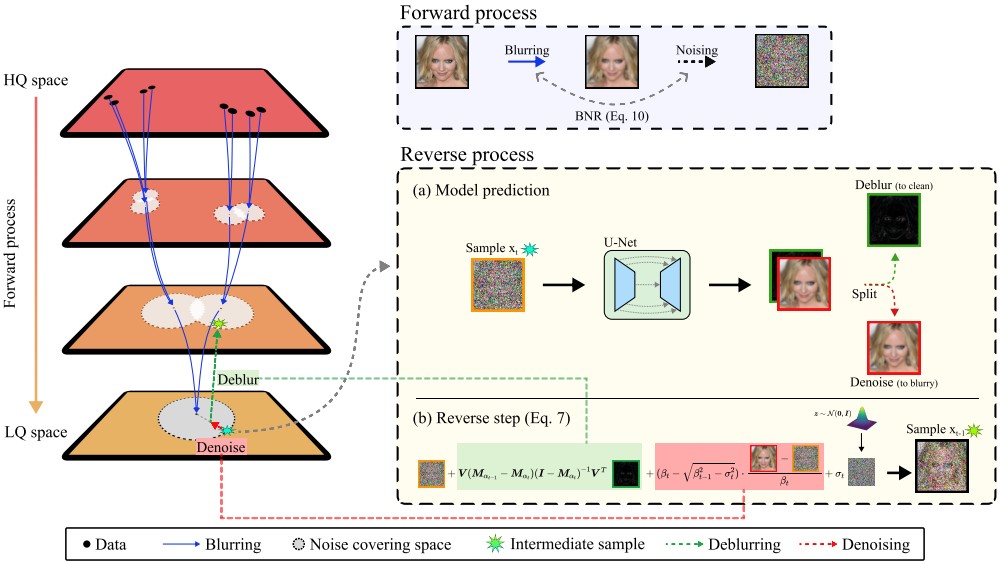

Figure 2: Workflow of the proposed diffusion process. The forward process progressively applies blurring and noise, controlled by the Blur-to-Noise Ratio (BNR), to degrade the sample from high quality to low quality. During this phase, training pairs are collected to train the prediction model (e.g., U-Net) for use in the reverse process. For sample generation, the reverse process works as follows: (a) The prediction model simultaneously performs denoising and deblurring. (b) With the prediction results, the reverse step transitions the sample from step $t$ to $t-1$. Specifically, the denoiser gradually guides the sample toward a blurry prediction, while the deblurring prediction helps return the sample to a higher-quality state.

factors $\alpha$ and $\beta$, which represent the blur and noise levels, respectively. In the forward process, the schedules of $\alpha$ and $\beta$ determine a blend of blurring and noising operations. Consequently, the reverse process iteratively recovers a high-quality image by deblurring and denoising a sampled image drawn from a prior distribution (i.e., a standard normal distribution). As shown in Fig. 2, mixing deblurring and denoising throughout the iterative image generation process distinguishes our approach from most existing diffusion models, which typically generate images through denoising alone. Moreover, our forward and generative processes are defined similarly to DDIM (Song et al., 2022), but with a focus on incorporating blur-noise operations into both stages. For brevity, we primarily address the key terms from DDIM (Song et al., 2022) that require adaptation for our model.

## 4.1 BLUR-NOISE FORWARD DIFFUSION PROCESSES

Our blur-noise forward process has the marginal distributions for $t \in \{1, \ldots, T\}$ as:

$$q(\boldsymbol{x}_{\alpha_t,\beta_t}|\boldsymbol{x}_0) = \mathcal{N}(\boldsymbol{V}\boldsymbol{M}_{\alpha_t}\boldsymbol{V}^T\boldsymbol{x}_0, \beta_t^2\boldsymbol{I}), \qquad (4)$$

where $\boldsymbol{V}^T$ and $\boldsymbol{V}$ denote the forward and inverse Discrete Cosine Transform (DCT), respectively. $\boldsymbol{M}_{\alpha_t}$, a diagonal matrix, specifies the Gaussian blurring mask in the DCT domain, which varies according to the Gaussian blur level $\alpha_t$. The parameter $\beta_t$ controls the level of Gaussian noise. The corruption sequences $\alpha_1, \ldots, \alpha_T$ and $\beta_1, \ldots, \beta_T$ are defined as monotonically increasing sequences. As with DDIM (Song et al., 2022), Eq. (4) requires the inference transition distributions (i.e., those used in the reverse process) $q_\sigma(.)$ for all $t > 1$ (see Appendix A.1) to be:

$$q_\sigma(\boldsymbol{x}_{\alpha_{t-1},\beta_{t-1}}|\boldsymbol{x}_{\alpha_t,\beta_t}, \boldsymbol{x}_0) = \mathcal{N}(\boldsymbol{V}\boldsymbol{M}_{\alpha_{t-1}}\boldsymbol{V}^T\boldsymbol{x}_0 + \sqrt{\beta_{t-1}^2 - \sigma_t^2} \cdot \frac{\boldsymbol{x}_{\alpha_t,\beta_t} - \boldsymbol{V}\boldsymbol{M}_{\alpha_t}\boldsymbol{V}^T\boldsymbol{x}_0}{\beta_t}, \sigma_t^2\boldsymbol{I}).$$

$$(5)$$

### 4.2 GENERATIVE PROCESS IN A DIVIDE-AND-CONQUER MANNER

Our generative process is a Markovian process specified by learnable transition distributions $p_\theta(\boldsymbol{x}_{\alpha_{t-1},\beta_{t-1}}|\boldsymbol{x}_{\alpha_t,\beta_t})$ for $t > 1$. The training of these distributions involves maximizing a variational lower bound that requires minimizing among others the sum of the KL-divergence terms $KL(q_\sigma(\boldsymbol{x}_{\alpha_{t-1},\beta_{t-1}}|\boldsymbol{x}_{\alpha_t,\beta_t},\boldsymbol{x}_0)\|p_\theta(\boldsymbol{x}_{\alpha_{t-1},\beta_{t-1}}|\boldsymbol{x}_{\alpha_t,\beta_t})$ for $t > 1$.

Notably, parameterizing $p_\theta(\boldsymbol{x}_{\alpha_{t-1},\beta_{t-1}}|\boldsymbol{x}_{\alpha_t,\beta_t})$ is to leverage the knowledge about $q_\sigma(\boldsymbol{x}_{\alpha_{t-1},\beta_{t-1}}|\boldsymbol{x}_{\alpha_t,\beta_t},\boldsymbol{x}_0)$ and learn a mean function to predict the mean in Eq. (5) based on the blurry-noisy observation $\boldsymbol{x}_{\alpha_t,\beta_t}$. At the inference time, we do not have access to the input $\boldsymbol{x}_0$; as such, one straightforward approach to parameterizing $p_\theta(\boldsymbol{x}_{\alpha_{t-1},\beta_{t-1}}|\boldsymbol{x}_{\alpha_t,\beta_t})$ is to learn a network $\hat{\boldsymbol{x}}_0 = F_\theta(\boldsymbol{x}_{\alpha_t,\beta_t}, t)$ that makes a prediction of $\boldsymbol{x}_0$ from $\boldsymbol{x}_{\alpha_t,\beta_t}$, and have $p_\theta(\boldsymbol{x}_{\alpha_{t-1},\beta_{t-1}}|\boldsymbol{x}_{\alpha_t,\beta_t}) = q_\sigma(\boldsymbol{x}_{\alpha_{t-1},\beta_{t-1}}|\boldsymbol{x}_{\alpha_t,\beta_t},\hat{\boldsymbol{x}}_0)$.

Instead of predicting $\boldsymbol{x}_0$ from $\boldsymbol{x}_{\alpha_t,\beta_t}$ directly with a single network, this work introduces a novel divide-and-conquer approach to parameterizing $p_\theta(\boldsymbol{x}_{\alpha_{t-1},\beta_{t-1}}|\boldsymbol{x}_{\alpha_t,\beta_t})$. This is motivated by the observation that the mean in Eq. (5) can be expressed alternatively as:

$$\boldsymbol{x}_{\alpha_t,\beta_t} + \underbrace{\boldsymbol{V}(\boldsymbol{M}_{\alpha_{t-1}} - \boldsymbol{M}_{\alpha_t})\boldsymbol{V}^T\boldsymbol{x}_0}_{\text{add missing high-frequency detail}} + (\beta_t - \sqrt{\beta_{t-1}^2 - \sigma_t^2}) \cdot \underbrace{\frac{\boldsymbol{V}\boldsymbol{M}_{\alpha_t}\boldsymbol{V}^T\boldsymbol{x}_0 - \boldsymbol{x}_{\alpha_t,\beta_t}}{\beta_t}}_{\text{direction pointing to blurry } \boldsymbol{x}_0}, \qquad (6)$$

where we have additionally added and subtracted the same term $\boldsymbol{x}_{\alpha_t,\beta_t} - \boldsymbol{V}\boldsymbol{M}_{\alpha_t}\boldsymbol{V}^T\boldsymbol{x}_0$. This alternative expression suggests that the task of parameterizing $p_\theta(\boldsymbol{x}_{\alpha_{t-1},\beta_{t-1}}|\boldsymbol{x}_{\alpha_t,\beta_t})$ by learning a mean function to predict that of Eq. (5) can be decomposed into two sub-tasks: deblurring and denoising. The former aims to reconstruct the high-frequency detail of $\boldsymbol{x}_0$ via the iterative generation process, while the latter is to recover a blurry version (i.e., $\boldsymbol{V}\boldsymbol{M}_{\alpha_t}\boldsymbol{V}^T\boldsymbol{x}_0$) of $\boldsymbol{x}_0$ from its noisy observation (i.e., $\boldsymbol{x}_{\alpha_t,\beta_t}$) by focusing on the reconstruction of the low-frequency components of $\boldsymbol{x}_0$. This interpretation allows us to learn two specialized networks for addressing these separate sub-tasks, leading to the more efficient and accurate generation of output images.

Specifically, we learn a network $D_\theta$ that takes the blurry-noisy observation $\boldsymbol{x}_{\alpha_t,\beta_t} = \boldsymbol{V}\boldsymbol{M}_{\alpha_t}\boldsymbol{V}^T\boldsymbol{x}_0 + \beta_t\boldsymbol{\epsilon}, \boldsymbol{\epsilon} \sim \mathcal{N}(\boldsymbol{0},\boldsymbol{I})$ as input to predict the noise-free yet blurry representation $\boldsymbol{V}\boldsymbol{M}_{\alpha_t}\boldsymbol{V}^T\boldsymbol{x}_0$ of $\boldsymbol{x}_0$. When learned successfully, $D_\theta$ is able to denoise $\boldsymbol{x}_{\alpha_t,\beta_t}$. For deblurring, a separate network $R_\theta$, which takes the same $\boldsymbol{x}_{\alpha_t,\beta_t}$ as input, is learned to update $\boldsymbol{V}\boldsymbol{M}_{\alpha_t}\boldsymbol{V}^T\boldsymbol{x}_0$ as $\boldsymbol{x}_0$. That is, $R_\theta$ aims to recover the missing high-frequency detail in $\boldsymbol{V}\boldsymbol{M}_{\alpha_t}\boldsymbol{V}^T\boldsymbol{x}_0$, in order to reconstruct $\boldsymbol{x}_0$. In symbols, $R_\theta$ is meant to predict $\boldsymbol{x}_{res_t} = \boldsymbol{x}_0 - \boldsymbol{V}\boldsymbol{M}_{\alpha_t}\boldsymbol{V}^T\boldsymbol{x}_0 = \boldsymbol{V}(\boldsymbol{I} - \boldsymbol{M}_{\alpha_t})\boldsymbol{V}^T\boldsymbol{x}_0$. With our proposed parameterization, and given that $\boldsymbol{M}_{\alpha_t}$ is a diagonal matrix, the mean function of $p_\theta(\boldsymbol{x}_{\alpha_{t-1},\beta_{t-1}}|\boldsymbol{x}_{\alpha_t,\beta_t})$ is:

$$\boldsymbol{x}_{\alpha_t,\beta_t} + \boldsymbol{V}(\boldsymbol{M}_{\alpha_{t-1}} - \boldsymbol{M}_{\alpha_t})(\boldsymbol{I} - \boldsymbol{M}_{\alpha_t})^{-1}\boldsymbol{V}^T R_\theta(\boldsymbol{x}_{\alpha_t,\beta_t},\alpha_t,\beta_t)$$
$$+ (\beta_t - \sqrt{\beta_{t-1}^2 - \sigma_t^2}) \cdot \frac{D_\theta(\boldsymbol{x}_{\alpha_t,\beta_t},\alpha_t,\beta_t) - \boldsymbol{x}_{\alpha_t,\beta_t}}{\beta_t}. \qquad (7)$$

Considering the equation $\boldsymbol{x}_{res_t} = \boldsymbol{V}(\boldsymbol{I} - \boldsymbol{M}_{\alpha_t})\boldsymbol{V}^T\boldsymbol{x}_0$, the second term $\boldsymbol{V}(\boldsymbol{M}_{\alpha_{t-1}} - \boldsymbol{M}_{\alpha_t})\boldsymbol{V}^T\boldsymbol{x}_0$ in Eq. (6) can be rewritten as $\boldsymbol{V}(\boldsymbol{M}_{\alpha_{t-1}} - \boldsymbol{M}_{\alpha_t})(\boldsymbol{I} - \boldsymbol{M}_{\alpha_t})^{-1}\boldsymbol{V}^T\boldsymbol{x}_{res_t}$ to match the form of its counterpart in Eq. (7). To approximate Eq. (6) using Eq. (7), we then train $D_\theta$ and $R_\theta$ by minimizing the following mean square error (MSE) losses

$$L(D_\theta) = \|D_\theta(\boldsymbol{x}_{\alpha_t,\beta_t},\alpha_t,\beta_t) - (\boldsymbol{V}\boldsymbol{M}_{\alpha_t}\boldsymbol{V}^T\boldsymbol{x}_0)\|_2^2, \text{ and} \qquad (8)$$

$$L(R_\theta) = \|R_\theta(\boldsymbol{x}_{\alpha_t,\beta_t},\alpha_t,\beta_t) - \boldsymbol{x}_{res_t}\|_2^2. \qquad (9)$$

Here, given a noisy and blurry observation $\boldsymbol{x}_{\alpha_t,\beta_t} = \boldsymbol{V}\boldsymbol{M}_{\alpha_t}\boldsymbol{V}^T\boldsymbol{x}_0 + \beta_t\boldsymbol{\epsilon}$ at time $t$, along with the corresponding corruption factors, $\alpha_t$ and $\beta_t$, $D_\theta$ (Denoiser) is trained to recover the pure blurry image by removing the noise component, while $R_\theta$ (Deblurrer) extracts the high-frequency detail $\boldsymbol{x}_{res_t}$. Finally, at time $t$, the prediction of the clean image is given by $\hat{\boldsymbol{x}}_0 = F_\theta = D_\theta + R_\theta$.

### 4.3 THE IMPACT OF BLUR-TO-NOISE RATIO ON MODEL BEHAVIOR AND DATA MANIFOLD

We have demonstrated how to combine two degradation factors —blurring and noising— into a unified diffusion process and introduced training objectives that simultaneously address deblurring and

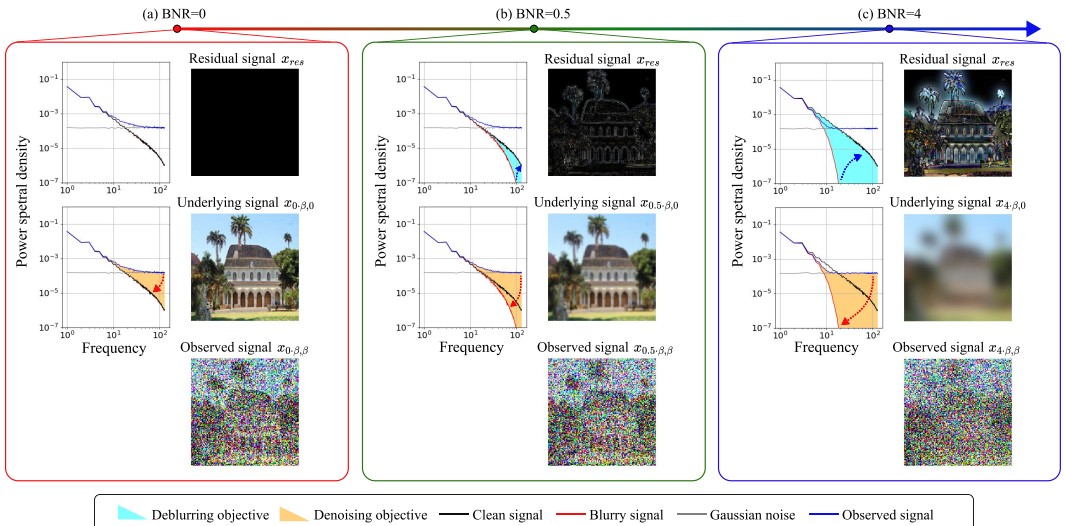

Figure 3: Impact of Varying BNRs on Model Behavior. We illustrate the observed signal, denoising target, and deblurring target, along with their respective signal spectrum analyses, across different BNR values. From left to right, the noise level remains constant while the BNR value increases. As the BNR rises, the denoising task (red arrow) becomes progressively easier, shifting more responsibility to the deblurring task (blue arrow) and effectively utilizing the spectral dependency of images. In contrast, when BNR = 0, the model requires a stronger denoiser to directly generate the image, without leveraging the spectral dependency assistance from the deblurrer.

denoising for image generation. However, the relationship between the blur and noise levels remains unclear. To explore this connection, we define a factor called the Blur-to-Noise Ratio (BNR):

$$BNR = \frac{Blur\ Level}{Noise\ Level} = \frac{\alpha}{\beta}, \tag{10}$$

which represents the ratio of the blur level to the noise level. As illustrated in Fig. 1, increasing the BNR value from 0 to $\infty$ transitions the diffusion path from hot to cold diffusion. To further investigate the model behavior with varying BNR values, we examine the learning objectives introduced in Sec. 4.2, which consist of two distinct branches targeting denoising and deblurring, respectively. Fig. 3 shows the signal spectra of images and the corresponding training objectives for different BNR values. With a constant noise level, an increase in BNR would raise the blur level, thereby simplifying the denoising task. This shift occurs because the denoiser no longer needs to restore high-frequency detail, transferring that responsibility to the deblurring task. Additionally, by effectively utilizing spectral dependency, the deblurrer can efficiently learn a mapping function from the low-frequency observation to its high-frequency counterpart.

In addition to affecting model behavior, varying BNR values also lead to shifts in data manifolds during forward iterations. To explore this in greater detail, we compare the data manifolds under low and high BNR scenarios with the same blur level in Fig. 4. At each forward or reverse step, low BNR cases exhibit a larger noise-covering space due to higher randomness compared to high BNR cases. Consequently, a high BNR results in reduced data diversity, making generated samples more likely to fall out of the data manifold, particularly during the early forward steps. When encountering out-of-manifold cases in sample generation, the diffusion network must handle degraded samples that were not seen during training, leading to less predictable outcomes and lower generation quality.

The analyses above reveal an inherent trade-off between model learning dynamics and data manifold shifts, where the choice of BNR value plays a crucial role in generation quality. As the diffusion process transitions from hot to cold, the model increasingly depends on leveraging the spectral dependency of images for learning. However, this shift also introduces the risk of divergence from the data manifold during generation, potentially resulting in degraded performance.

## 4.4 SELECTING BNR

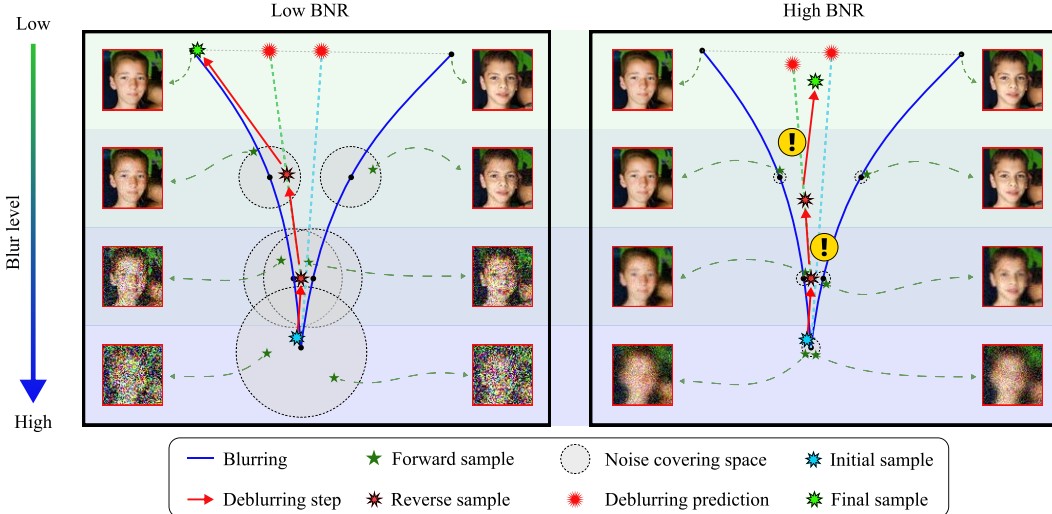

Figure 4: Illustration of the connection between BNR and the data manifold. When comparing two different BNRs at the same blur level, a higher BNR corresponds to a smaller noise scale, resulting in a narrower noise-covering space, as shown on the right. In the deblurring (reverse) step, a sample is guided toward the deblurring target, representing the mean image of all possible paired outputs. It is important to note that during the forward process, a single low-quality (LQ) sample is typically paired with multiple high-quality (HQ) samples for training. Due to this ill-posed nature of the deblurring task, samples with higher BNR values are more likely to deviate from the data manifold during the transition. Once samples fall out of the manifold, the neural network struggles to produce accurate predictions, leading to a decline in generation quality.

We've discussed how BNR influences model behavior and the data manifold. A higher BNR allows the model to better exploit spectral dependency, simplifying learning by shifting the focus to de-blurring. However, it also heightens the risk of samples deviating from the data manifold. Striking the right balance is crucial, as one must weigh the benefits of more straightforward neural network training against the risk of such deviations. This raises the question of whether a better BNR value exists that preserves the data manifold's integrity while enhancing model training.

An interesting observation from previous studies (van der Schaaf & van Hateren, 1996; Rissanen et al., 2023) is that the power spectral density of natural images follows an approximate power law, $1/f^\alpha$, where $\alpha \approx 2$. In contrast, Gaussian white noise exhibits a flat frequency response across all frequency bands. This discrepancy results in a much lower signal-to-noise ratio (SNR) in high-frequency bands compared to low-frequency ones. When the SNR in these bands is sufficiently low, the observed signal becomes dominated by noise, helping to maintain the integrity of the data manifold while attenuating the image signal in these bands.

As shown in Fig. 3(b), our empirical findings indicate that selecting **BNR=0.5** causes the image signal to begin attenuating when noise intensity exceeds the signal in these frequency bands, keeping the blurry-noisy signal comparable to the noisy signal in hot diffusion. Beyond this threshold, Fig. 3(c) illustrates that for a higher BNR value, the observed signal diverges from that in hot diffusion, as the image signal attenuates significantly before noise dominates those frequency bands.

# 5 EXPERIMENTS

## 5.1 IMAGE GENERATION

**Datasets.** We validate our proposed diffusion process on three widely used benchmarks: CIFAR-10 (Krizhevsky, 2009) $32 \times 32$, FFHQ (Karras et al., 2019) $64 \times 64$, and LSUN-church (Yu et al., 2016) $128 \times 128$. These datasets were chosen to demonstrate the effectiveness of our method across various scenarios. The CIFAR-10 dataset contains $32 \times 32$ color images across 10 classes, allowing us to

Table 1: Quantitative results and comparison for $32 \times 32$ and $64 \times 64$ image generation tasks on CIFAR-10 (Krizhevsky, 2009) and FFHQ (Karras et al., 2019) datasets correspondingly. Lower FID and higher IS scores indicate better sample quality. NFE denotes the **"Number of Function Evaluations"**. The best results are highlighted in bold; the second-best results are underlined.

| Methods | NFE ↓ | FID ↓ | IS ↑ |
|---|---|---|---|
| Unconditional CIFAR-10 | | | |
| Cold Diffusion (Blur) (Bansal et al., 2022) | 50 | 80.08 | - |
| IHDM (Rissanen et al., 2023) | 200 | 18.96 | - |
| Blurring Diffusion (Hoogeboom & Salimans, 2024) | 1000 | 3.17 | 9.51 |
| EDM (Karras et al., 2022) | 35 | 1.97 | 9.78 |
| EDM-ES (Ning et al., 2024) | 35 | 1.95 | - |
| STF (Xu et al., 2023b) | 35 | 1.92 | 9.79 |
| PFGM++ (Xu et al., 2023a) | 35 | 1.91 | - |
| Ours | 35 | **1.85** | **10.02** |
| Class-conditional CIFAR-10 | | | |
| EDM (Karras et al., 2022) | 35 | 1.79 | - |
| EDM-ES (Ning et al., 2024) | 35 | 1.80 | - |
| PFGM++ (Xu et al., 2023a) | 35 | 1.74 | - |
| Ours | 35 | **1.68** | **10.19** |
| FFHQ $64 \times 64$ | | | |
| EDM (Karras et al., 2022) | 79 | 2.53 | - |
| PFGM++ (Xu et al., 2023a) | 79 | 2.43 | - |
| Ours | 79 | **2.29** | **3.41** |

Table 2: Quantitative results and comparisons for $128 \times 128$ image generation tasks on the unconditional LSUN-church (Yu et al., 2016) dataset. For a fair comparison, we evaluate sample quality using the same number of samples as in previous studies.

| Methods | NFE ↓ | FID ↓ |
|---|---|---|
| Number of samples = 10k | | |
| Denoising Diffusion (Hoogeboom & Salimans, 2024) | 1000 | 4.68 |
| Blurring Diffusion (Hoogeboom & Salimans, 2024) | 1000 | 3.88 |
| Ours | 511 | 3.47 |
| Number of samples = 50k | | |
| IHDM (Rissanen et al., 2023) | 400 | 45.06 |
| Ours | 511 | 2.56 |

Table 3: Ablation study on the impact of different BNR values for CIFAR-10, with a fixed number of sampling steps (NFE=35).

| BNR | FID ↓ | IS ↑ |
|---|---|---|
| 0 (EDM (Karras et al., 2022)) | 1.97 | 9.78 |
| 0.1 | 1.97 | 9.96 |
| 0.3 | 1.90 | 10.02 |
| 0.5 | 1.85 | 10.02 |
| 0.65 | 1.91 | 10.00 |
| 1 | 2.01 | 9.96 |
| 2 | 2.57 | 9.89 |
| 10 | 11.97 | 8.51 |

test both unconditional and class-conditional image generation. For the FFHQ and LSUN-church datasets, we evaluate the model in unconditional settings. The FFHQ $64 \times 64$ dataset comprises 70,000 images of human faces, which have a higher degree of shared structure compared to general scene datasets. The LSUN-church $128 \times 128$ dataset features images of church scenes, enabling us to validate our method in higher-resolution scenarios.

**Implementation Details.** For training, we adopt the improved DDPM++/NCSN++ (Song et al., 2021) network architectures, training strategies, and hyperparameters from the state-of-the-art diffusion model, EDM (Karras et al., 2022). Modifications are made to enable the network to accept two conditioning signals—the blur and noise levels—and we double the output channels to produce predictors for deblurring and denoising, respectively. In our current design, the two networks share most components, so altering the BNR keeps the model capacity nearly constant, ensuring a fair comparison. For sampling, we adapt Heun's $2^{nd}$ solver, following EDM (Karras et al., 2022). More details are available in Appendix B.

**Performance Comparison.** To evaluate image generation quality, we use two commonly adopted benchmarks: Fréchet Inception Distance (FID) (Heusel et al., 2018) and Inception Score (IS) (Salimans et al., 2016). The FID score measures the distance between the generated and reference datasets; a lower FID score indicates greater similarity between the two, reflecting better recovery of the data distribution by the generative model. The Inception Score is computed by passing generated images through a pre-trained classifier. The optimal IS is achieved when the entropy of the label distribution for the generated images is minimized and the predictions are evenly spread across classes, indicating sharp and diverse generated images.

Following established procedures, we sample 50,000 images over three rounds and report the minimum scores to mitigate random variation effects. As shown in Tab. 1, we assess sample quality using FID and IS alongside the number of function evaluations (NFE) during sampling—a metric closely related to the sampling speed of diffusion-based methods. Our approach significantly enhances the performance of the baseline model, EDM, across both CIFAR-10 and FFHQ datasets, regardless of their differing characteristics. This improvement is evident in both conditional and unconditional settings on CIFAR-10, outperforming recent methods designed to enhance EDM.

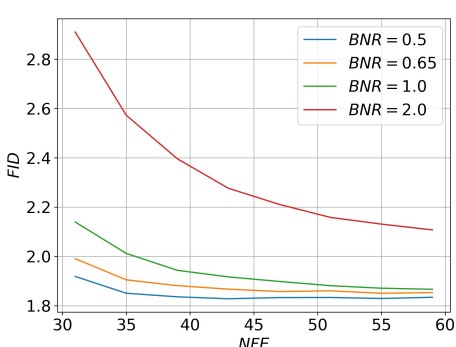
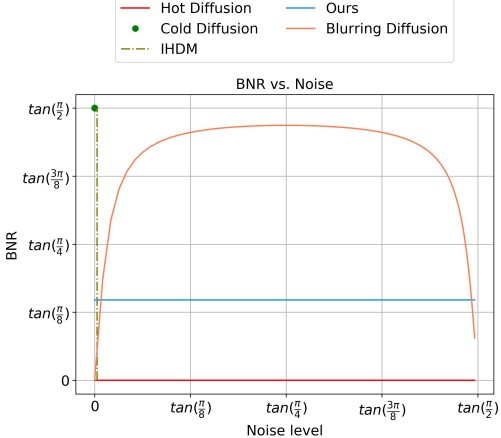

Figure 5: Illustration of sample quality corresponding to different BNR and NFE. Each curve represents a specific BNR value. As shown in the chart, higher BNR values require more sampling steps to achieve better sample quality.

Figure 6: Comparison of blur and noise schedules with previous methods. Most related studies, except the hot diffusion, selected higher BNR values in their schedules, leading to out-of-manifold issues. Details in Appendix A.2.

Moreover, in addition to comparing our approach with techniques aimed at improving hot diffusion, we evaluate it against Cold Diffusion and intermediate methods such as IHDM (Rissanen et al., 2023) and Blurring Diffusion (Hoogeboom & Salimans, 2024). Our method not only achieves superior sample quality but also requires significantly fewer sampling steps for image generation. In Tab. 2, we also show our method outperforms the previous methods on a more complex and higher-resolution dataset, LSUN-church $128 \times 128$. To make a fair comparison, we use the same number of samples for the FID evaluation since the FID score is sensitive to the number of samples.

## 5.2 ANALYSIS OF DIFFERENT BNRS AND SCHEDULING

**Relation between BNRs and Data Manifolds.** We first validate our assumptions by testing various BNR values ranging from 0 to 10. As shown in Tab. 3, using the same number of sampling steps as the baseline method, EDM (Karras et al., 2022), the sample quality improves as the BNR value increases up to 0.5. Beyond this threshold, performance rapidly declines, eventually falling below the baseline. These results support our hypothesis in Sec. 4.4 that a BNR value of 0.5, guided by spectral analysis, can effectively balance the trade-off between mitigating negative impacts on data manifold shifts and enhancing model learning. As discussed in Sec. 4.4, when the BNR value exceeds 0.5, the data manifold tends to shift significantly. Consequently, following the same sampling steps, the samples are more likely to fall outside the manifold, resulting in lower generation quality. This suggests that higher BNR values may require additional sampling steps to mitigate out-of-manifold issues and enhance sample quality. In Fig. 5, we further assess the impact of different BNR values and sampling steps on sample quality. The results confirm that higher BNR values indeed demand more sampling steps to reach better sample quality, supporting our analysis.

**Revisiting the BNR Scheduling in Prior Studies.** We conduct experiments to compare the BNR schedules from prior studies with our proposed method. Specifically, we re-implement the BNR schedule from Blurring Diffusion (Hoogeboom & Salimans, 2024) under the same experimental conditions to eliminate the effects of differing parameterization techniques. The results, presented in Tab. 4, indicate that Blurring Diffusion's BNR schedule results in poor generation quality when using fewer sampling steps, although quality improves significantly with more steps. This highlights a limitation in Blurring Diffusion's BNR scheduling, as depicted in Fig. 6, where higher BNR values necessitate additional sampling steps to prevent the reverse process from deviating from the data manifold. Consequently, fewer steps result in poorer sample quality. These findings help explain why previous approaches, such as cool diffusion, have struggled to generate high-quality samples, particularly under limited sampling budgets.

Table 4: We re-implement Blurring Diffusion using our parameterization and training scheme on CIFAR-10. Results marked with $^*$ are those reported by Hoogeboom & Salimans (2024).

| BNR Schedule | NFE ↓ | FID ↓ | IS ↑ |
|---|---|---|---|
| | 35 | 12.97 | 8.57 |
| | 79 | 4.13 | 9.27 |
| | 159 | 2.91 | 9.46 |
| Blurring Diffusion (Hoogeboom & Salimans, 2024) | 239 | 2.77 | 9.52 |
| | 319 | 2.73 | 9.54 |
| | 399 | 2.71 | 9.54 |
| | 999 | 2.68 | 9.56 |
| Blurring Diffusion$^*$ (Hoogeboom & Salimans, 2024) | 1000 | 3.17 | 9.51 |

Table 5: An ablation study on different parameterization of $\boldsymbol{x}_0$ and $\boldsymbol{V}\boldsymbol{M}_{\alpha_t}\boldsymbol{V}^T\boldsymbol{x}_0$ in Eq. (6). We fix a constant number of sampling steps (NFE=35) to investigate various parameterizations for the generation task using CIFAR-10.

| | Training Objectives | | Parameterization of | | FID ↓ |
|---|---|---|---|---|---|
| | $R_\theta$ | $D_\theta$ | $\boldsymbol{x}_0$ | $\boldsymbol{V}\boldsymbol{M}_{\alpha_t}\boldsymbol{V}^T\boldsymbol{x}_0$ | |
| (a) | $\boldsymbol{x}_0$ | - | $R_\theta$ | $\boldsymbol{V}\boldsymbol{M}_{\alpha_t}\boldsymbol{V}^T R_\theta$ | 13.19 |
| (b) | - | $\boldsymbol{V}\boldsymbol{M}_{\alpha_t}\boldsymbol{V}^T\boldsymbol{x}_0$ | $\boldsymbol{V}(\boldsymbol{M}_{\alpha_t})^{-1}\boldsymbol{V}^T D_\theta$ | $D_\theta$ | 9.09 |
| (c) | $\boldsymbol{x}_0$ | $\boldsymbol{V}\boldsymbol{M}_{\alpha_t}\boldsymbol{V}^T\boldsymbol{x}_0$ | $R_\theta$ | $D_\theta$ | 1.98 |
| (d) | $\boldsymbol{x}_{res_t}$ | $\boldsymbol{V}\boldsymbol{M}_{\alpha_t}\boldsymbol{V}^T\boldsymbol{x}_0$ | $\boldsymbol{V}(\boldsymbol{I}-\boldsymbol{M}_{\alpha_t})^{-1}\boldsymbol{V}^T R_\theta$ | $D_\theta$ | 1.85 |

## 5.3 ABLATION STUDIES ON TRAINING OBJECTIVES AND VARIATIONS OF PARAMETERIZATIONS

In Sec. 4.2, we reformulate the reverse function, Eq. (6), to simplify the neural network's training objectives through a divide-and-conquer approach. This method separates the task of predicting the clean signal $\boldsymbol{x}_0$ into two sub-tasks: denoising to a blurry signal and predicting the residual signal for deblurring. In this subsection, we explore various parameterization strategies derived from Eq. (6) and evaluate their performances, demonstrating the advantages of our divide-and-conquer strategy for model learning.

As shown in Tab. 5(a), using a single branch to directly predict the entire clean signal results in significantly poorer generation quality, as this task proves too challenging. In Tab. 5(b), shifting the target to learn a blurry signal yields a slight improvement in sample quality since this target is easier to model; however, it still faces issues with inaccuracies in high-frequency components, which may be exacerbated during sampling, occasionally resulting to noisy patterns in the generated samples. In Tab. 5(c), employing two branches to predict both clean and blurry signals effectively addresses these challenges, leading to substantially better results, though they remain comparable to those of hot diffusion models as EDM (Karras et al., 2022). Finally, in Tab. 5(d), the proposed divide-and-conquer strategy further improves performance, benefiting especially from the BNR schedule and the parameterization. More visual examples are provided in Fig. 11.

## 6 CONCLUSION

In this paper, we introduce a unified Warm Diffusion framework that effectively bridges the gap between hot and cold diffusion models while addressing their inherent limitations. Our analysis reveals that hot diffusion models underutilize the spectral dependency of images, whereas cold diffusion models risk reverse sampling steps that deviate from the data manifolds. By examining the Blur-to-Noise Ratio (BNR), we uncover its significant influence on model behavior and data manifolds. This insight enables us to propose a strategy for balancing the trade-off between hot and cold diffusion, ultimately enhancing diffusion models for image generation. Experimental results across various benchmarks validate the effectiveness of our approach, demonstrating improvements in sample quality over state-of-the-art diffusion models.

ACKNOWLEDGMENTS

This work was financially supported in part (project number: 112UA10019) by the Co-creation Platform of the Industry Academia Innovation School, NYCU, under the framework of the National Key Fields Industry-University Cooperation and Skilled Personnel Training Act, from the Ministry of Education (MOE) and industry partners in Taiwan. It also supported in part by the National Science and Technology Council, Taiwan, under Grant NSTC-112-2221-E-A49-089-MY3, Grant NSTC-110-2221-E-A49-066-MY3, Grant NSTC-111-2634-F-A49-010, Grant NSTC-112-2425-H-A49-001, and in part by the Higher Education Sprout Project of the National Yang Ming Chiao Tung University and the Ministry of Education (MOE), Taiwan. We also would like to express our gratitude for the support from MediaTek Inc, Hon Hai Research Institute (HHRI), E.SUN Financial Holding Co Ltd, Advantech Co Ltd, Industrial Technology Research Institute (ITRI)

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

## A  DERIVATION

### A.1  PROOF

In this section, we prove that the inference transition distribution defined in Eq. (5) matches the marginal distribution defined in Eq. (4), using an induction hypothesis.

**Base Case:**

For $t = T$, it is given that

$$q_\sigma(\boldsymbol{x}_{\alpha_T,\beta_T}|\boldsymbol{x}_0) = \mathcal{N}(\boldsymbol{V}\boldsymbol{M}_{\alpha_T}\boldsymbol{V}^T\boldsymbol{x}_0, \beta_T^2\boldsymbol{I}), \tag{11}$$

so the base case holds.

**Induction Hypothesis:**

Assume that for $t$, the following holds

$$q_\sigma(\boldsymbol{x}_{\alpha_t,\beta_t}|\boldsymbol{x}_0) = \mathcal{N}(\boldsymbol{V}\boldsymbol{M}_{\alpha_t}\boldsymbol{V}^T\boldsymbol{x}_0, \beta_t^2\boldsymbol{I}). \tag{12}$$

We now aim to show that it holds for $t - 1$.

**Inductive Step:**

We have

$$q_\sigma(\boldsymbol{x}_{\alpha_{t-1},\beta_{t-1}}|\boldsymbol{x}_0) = \int_{\boldsymbol{x}_{\alpha_t,\beta_t}} q_\sigma(\boldsymbol{x}_{\alpha_t,\beta_t}|\boldsymbol{x}_0)q_\sigma(\boldsymbol{x}_{\alpha_{t-1},\beta_{t-1}}|\boldsymbol{x}_{\alpha_t,\beta_t},\boldsymbol{x}_0)d\boldsymbol{x}_{\alpha_t,\beta_t} \tag{13}$$

and also the inference transition distribution

$$q_\sigma(\boldsymbol{x}_{\alpha_{t-1},\beta_{t-1}}|\boldsymbol{x}_{\alpha_t,\beta_t},\boldsymbol{x}_0) = \mathcal{N}(\boldsymbol{V}\boldsymbol{M}_{\alpha_{t-1}}\boldsymbol{V}^T\boldsymbol{x}_0 + \sqrt{\beta_{t-1}^2 - \sigma_t^2}(\frac{\boldsymbol{x}_{\alpha_t,\beta_t} - \boldsymbol{V}\boldsymbol{M}_{\alpha_t}\boldsymbol{V}^T\boldsymbol{x}_0}{\beta_t}), \sigma_t^2\boldsymbol{I}). \tag{14}$$

Following Bishop & Nasrabadi (2006) (2.115), we have that $q_\sigma(\boldsymbol{x}_{\alpha_{t-1},\beta_{t-1}}|\boldsymbol{x}_0)$ is Gaussian, denoted as $\mathcal{N}(\boldsymbol{\mu}_{t-1}, \boldsymbol{\Sigma}_{t-1})$ where

$$\begin{aligned}
\boldsymbol{\mu}_{t-1} &= \boldsymbol{V}\boldsymbol{M}_{\alpha_{t-1}}\boldsymbol{V}^T\boldsymbol{x}_0 + \sqrt{\beta_{t-1}^2 - \sigma_t^2}(\frac{\boldsymbol{V}\boldsymbol{M}_{\alpha_t}\boldsymbol{V}^T\boldsymbol{x}_0 - \boldsymbol{V}\boldsymbol{M}_{\alpha_t}\boldsymbol{V}^T\boldsymbol{x}_0}{\beta_t}) \\
&= \boldsymbol{V}\boldsymbol{M}_{\alpha_{t-1}}\boldsymbol{V}^T\boldsymbol{x}_0
\end{aligned} \tag{15}$$

and

$$\boldsymbol{\Sigma}_{t-1} = \sigma_t^2\boldsymbol{I} + (\frac{\beta_{t-1}^2 - \sigma_t^2}{\beta_t^2})\beta_t^2\boldsymbol{I} = \beta_{t-1}^2\boldsymbol{I}. \tag{16}$$

Therefore we have

$$q_\sigma(\boldsymbol{x}_{\alpha_{t-1},\beta_{t-1}}|\boldsymbol{x}_0) = \mathcal{N}(\boldsymbol{V}\boldsymbol{M}_{\alpha_{t-1}}\boldsymbol{V}^T\boldsymbol{x}_0, \beta_{t-1}^2\boldsymbol{I}). \tag{17}$$

**Conclusion:**

The marginal distribution $q_\sigma(\boldsymbol{x}_{\alpha_t,\beta_t}|\boldsymbol{x}_0) = \mathcal{N}(\boldsymbol{V}\boldsymbol{M}_{\alpha_t}\boldsymbol{V}^T\boldsymbol{x}_0, \beta_t^2\boldsymbol{I})$ holds for all t according to the derivation above.

## A.2 BNR Schedule of Prior Studies

Here, we further analyze the BNR schedule of prior studies as depicted in Fig. 6. For Blurring Diffusion (Hoogeboom & Salimans, 2024), we begin by examining its blurring and noising schedule indexed by $t \in \{1, 2, \ldots, T\}$. As defined in (Hoogeboom & Salimans, 2024), the blurring schedule, $\alpha_t$, is parameterized as:

$$\alpha_t = 20\sin^2(\frac{\pi t}{2T}). \tag{18}$$

Given that a cosine noise schedule is applied—scaling the signal throughout the diffusion process—the noising schedule, $\beta_t$, can be parameterized from the perspective of the signal-to-noise ratio (SNR) as:

$$\beta_t \approx \frac{\sin(\frac{\pi t}{2T})}{\cos(\frac{\pi t}{2T})} = \tan(\frac{\pi t}{2T}). \tag{19}$$

Thus the BNR schedule can be written as:

$$\begin{aligned}
BNR_t = \frac{\alpha_t}{\beta_t} &= \frac{20\sin^2(\frac{\pi t}{2T})}{\tan(\frac{\pi t}{2T})} \\
&= 20\sin(\frac{\pi t}{2T})\cos(\frac{\pi t}{2T}) \\
&= 10\sin(\frac{\pi t}{T}).
\end{aligned} \tag{20}$$

Regarding other related work, IHDM (Rissanen et al., 2023) introduces a diffusion process by applying a small, constant amount of noise while progressively increasing blur levels. As a result, the BNR schedule for IHDM is represented as a **vertical line** in Fig. 6. In contrast, Bansal et al. (2022) develop Cold Diffusion by constructing the diffusion process entirely without noise, leading to a BNR value of $\infty$.

## B Additional Details of our Implementation

We adopt the network architectures, training techniques, and hyperparameters from the state-of-the-art diffusion model, EDM (Karras et al., 2022), making only minor modifications to preserve constant model capacity. In this section, we first provide detailed information on the architecture and training settings, **demonstrating that the observed improvement in generation quality is not due to a larger or more complex network design nor to hyperparameter fine-tuning**. Subsequently, we present the algorithms for both training and sampling within our proposed diffusion process to facilitate a clearer understanding of the methodology.

### B.1 Network Architectures and Hyperparameters

In Tab. 6, we list the hyperparameters used in our experiments. For CIFAR-10 and FFHQ, we adopt the same settings as EDM (Karras et al., 2022), without tuning for optimal hyperparameters. For LSUN-church, we follow the network architecture and settings from Blurring Diffusion to ensure a fair comparison. Across all datasets, we apply slight modifications to the network architecture, as depicted in Fig. 7. These modifications include: (1) incorporating an additional conditioning signal for the blurring level, using an embedding branch fused with the noise-level embedding, and (2) doubling the output channels of the neural network and splitting them to compute separate losses, allowing the model to predict both the residual high-frequency detail and the underlying blurry signal. These two changes result in less than a $0.15\,\%_o$ increase in model size, as shown in Tab. 6, indicating that the observed improvement in sample quality stems from the proposed method rather than the architecture.

Table 6: Hyperparameters and model sizes used in the experiments.

| Hyperparameter | CIFAR-10 | | FFHQ $64 \times 64$ | | LSUN-church $128 \times 128$ | |
|---|---|---|---|---|---|---|
| | Baseline | Ours | Baseline | Ours | Baseline | Ours |
| Number of GPUs | 8 | 8 | 8 | 8 | 8 | 8 |
| Duration (Mimg) | 200 | 200 | 200 | 200 | 512 | 200 |
| Minibatch size | 512 | 512 | 256 | 256 | 256 | 256 |
| Learning rate $\times 10^{-4}$ | 10 | 10 | 2 | 2 | 1 | 1 |
| LR ramp-up (Mimg) | 10 | 10 | 10 | 10 | 10 | 10 |
| EMA half-life (Mimg) | 0.5 | 0.5 | 0.5 | 0.5 | 0.89 | 0.5 |
| Dropout probability | 0.13 | 0.13 | 0.05 | 0.05 | 0.1 | 0.1 |
| Channel multiplier | 128 | 128 | 128 | 128 | 64 | 64 |
| Channels per resolution | 2-2-2 | 2-2-2 | 1-2-2-2 | 1-2-2-2 | 1-2-4-6-8 | 1-2-4-6-8 |
| Residual blocks per resolution | 4 | 4 | 4 | 4 | 3 | 3 |
| Attention resolutions | $\{16\}$ | $\{16\}$ | $\{16\}$ | $\{16\}$ | $\{8, 16, 32\}$ | $\{8, 16, 32\}$ |
| Attention heads | 1 | 1 | 1 | 1 | 4-6-8 | 4-6-8 |
| Model size | 55.734 M | 55.741 M | 62.761 M | 62.765 M | 117.955 M | 117.957 M |

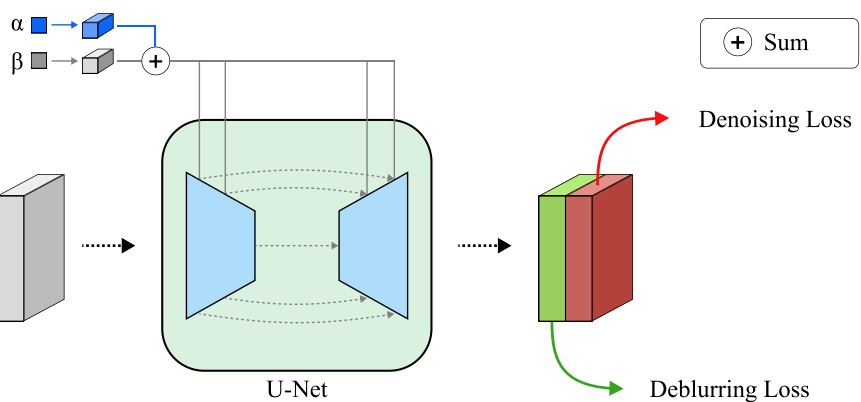

Figure 7: Our model architecture includes an additional embedding branch to incorporate a conditioning signal for the blurring and noising levels. Furthermore, to enable the network to simultaneously perform both deblurring and denoising, the output channels are doubled and split to handle each task separately.

## B.2 PROCEDURE OF TRAINING AND SAMPLING

We present the training procedure in Algorithm 1, which follows the training scheme of the state-of-the-art diffusion model EDM (Karras et al., 2022), with slight modifications. To prepare the training data pairs, after sampling a noise level $\beta$, we further determine a corresponding blur level $\alpha$ based on the predefined BNR value. The image signals are then transformed into blurry and noisy signals according to the sampled blur and noise levels. The decoding neural network is then trained to simultaneously denoise and deblur these signals, conditioned on the blur and noise levels. Unlike methods that require two separate function evaluations for denoising and deblurring, we utilize a single forward pass to predict both signals at once and split the output into two branches, each handling one task.

In Algorithm 2, we outline the sampling procedure of our proposed diffusion process, integrated with Heun's $2^{nd}$ order sampling method, similar to EDM (Karras et al., 2022). Following a predefined sampling schedule, for each reverse step, we first use the denoiser's prediction to guide the sample toward a blurry prediction, then apply the deblurring prediction to move the sample toward a sharper state, as outlined in Eq. (6). To integrate Heun's $2^{nd}$ order method, we temporarily update the signal $x_i$ to obtain $x_{i-1}$, then repeat the process to obtain a refined update direction. The two predictions from consecutive iterations are averaged to correct the update step, yielding the final sample $x_{i-1}$. This process is applied throughout, except for the last step, ensuring more accurate and stable sampling.

---

**Algorithm 1** Training Phase

---

1: **Require:** Hyperparameters $\{P_{mean}, P_{std}, BNR\}$
2: **Initialize** Neural network $F_\theta$
3: **repeat**
4:    $\boldsymbol{x}_0 \sim q(\boldsymbol{x}_0)$                                    ▷ Sample from training dataset
5:    $\boldsymbol{\epsilon} \sim \mathcal{N}(\boldsymbol{0}, \boldsymbol{I})$                                 ▷ Sample a Gaussian noise
6:    $\ln(\beta) \sim \mathcal{N}(P_{mean}, P_{std}^2)$                      ▷ Sample a noise level
7:    $\alpha = BNR \times \beta$             ▷ Get corresponding blur level from given noise level
8:    $\boldsymbol{x}_{\alpha,0} = \boldsymbol{V}\boldsymbol{M}_\alpha\boldsymbol{V}^T\boldsymbol{x}_0$                      ▷ Apply blurring operation
9:    $\boldsymbol{x}_{\alpha,\beta} = \boldsymbol{x}_{\alpha,0} + \beta\boldsymbol{\epsilon}$                                ▷ Add noise
10:   $\boldsymbol{x}_{res} = \boldsymbol{x}_0 - \boldsymbol{x}_{\alpha,0}$                       ▷ Compute the residual signal
11:   $\hat{D}_\theta, \hat{R}_\theta = F_\theta(\boldsymbol{x}_{\alpha,\beta}; \alpha, \beta)$              ▷ Split the output of the neural network
12:   Take gradient step on
13:      $\nabla_\theta \lambda(\beta)(\|\hat{D}_\theta - \boldsymbol{x}_{\alpha,0}\|^2 + \|\hat{R}_\theta - \boldsymbol{x}_{res}\|^2)$    ▷ Jointly learn denoising and deblurring
14: **until** converged

---

**Algorithm 2** Generation phase: Deterministic sampling with Heun's $2^{nd}$ order method

---

1: **Require:** Neural network $F_\theta$, Sampling schedule $\{(\alpha_0, \beta_0), (\alpha_1, \beta_1), \ldots, (\alpha_N, \beta_N)\}$
2: **sample** $\boldsymbol{x}_N \sim \mathcal{N}(\boldsymbol{0}, \beta_N^2\boldsymbol{I})$
3: **for** $i \in \{N, N-1, \ldots, 1\}$ **do**
4:    $\hat{D}_\theta, \hat{R}_\theta = F_\theta(\boldsymbol{x}_i; \alpha_i, \beta_i)$                       ▷ Split the model prediction
5:    $\hat{\boldsymbol{\epsilon}} = \frac{\boldsymbol{x}_i - \hat{D}_\theta}{\beta_i}$                                  ▷ First-order gradient term
6:    $\hat{\boldsymbol{x}}_0 = \boldsymbol{V}(\boldsymbol{I} - \boldsymbol{M}_{\alpha_i})^{-1}\boldsymbol{V}^T\hat{R}_\theta$        ▷ Transform residual signal $\boldsymbol{x}_{res}$ to get $\boldsymbol{x}_0$
7:    $\boldsymbol{x}_{i-1} = \boldsymbol{x}_i + V(\boldsymbol{M}_{\alpha_{i-1}} - \boldsymbol{M}_{\alpha_i})\boldsymbol{V}^T\hat{\boldsymbol{x}}_0 + (\beta_{i-1} - \beta_i)\hat{\boldsymbol{\epsilon}}$   ▷ Eq. (6), taking a Euler's step

8:    **if** $i \neq 1$ **then**                 ▷ Apply second-order correction except for the last step
9:       $\hat{D}'_\theta, \hat{R}'_\theta = F_\theta(\boldsymbol{x}_{i-1}; \alpha_{i-1}, \beta_{i-1})$
10:      $\hat{\boldsymbol{\epsilon}}' = \frac{\boldsymbol{x}_{i-1} - \hat{D}'_\theta}{\beta_{i-1}}$
11:      $\hat{\boldsymbol{x}}'_0 = V(\boldsymbol{I} - \boldsymbol{M}_{\alpha_{i-1}})^{-1}\boldsymbol{V}^T\hat{R}'_\theta$
12:      $\boldsymbol{x}_{i-1} = \boldsymbol{x}_i + \boldsymbol{V}(\boldsymbol{M}_{\alpha_{i-1}} - \boldsymbol{M}_{\alpha_i})\boldsymbol{V}^T(\frac{\hat{\boldsymbol{x}}_0 + \hat{\boldsymbol{x}}'_0}{2}) + (\beta_{i-1} - \beta_i)(\frac{\hat{\boldsymbol{\epsilon}} + \hat{\boldsymbol{\epsilon}}'}{2})$
13:    **end if**
14: **end for**
15: **return** $\boldsymbol{x}_0$

---

## C   ADDITIONAL EXPERIMENTS

### C.1   SAMPLING SCHEDULES FOR IMAGE GENERATION

This section explains how the noise level schedule affects **the reverse process** in our diffusion model. Before starting the reverse process, a predefined sequence of blur and noise levels is needed for each reverse step. Since the blur level in our method is controlled by the BNR and linked to the noise level, we only need to focus on the schedule of the noise level.

Following the formula proposed by Karras et al. (2022), the sequence of noise levels can be formulated as:

$$\beta_{0 < i \leq N} = (\beta_{max}^{\frac{1}{\rho}} + \frac{N-i}{N-1}(\beta_{min}^{\frac{1}{\rho}} - \beta_{max}^{\frac{1}{\rho}}))^\rho, \beta_0 = 0, \tag{21}$$

where $\rho$ controls the emphasis on different phases of noise levels. Setting $\rho = 1$ corresponds to uniform discretization, while a higher $\rho$ places more emphasis on lower noise levels, resulting in more sampling steps in this phase.

Empirically, Karras et al. (2022) found that $\rho = 7$ worked well across various tasks, and adopted this value in all experiments. However, we discovered that our method benefits from using a higher $\rho$ value, as illustrated in Fig. 8. This improvement arises because our model learns a simpler target in the early stages of the reverse diffusion process, thereby reducing prediction error. By shifting the focus to lower noise levels, more sampling

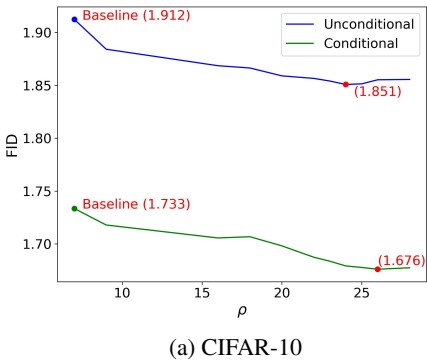
(a) CIFAR-10

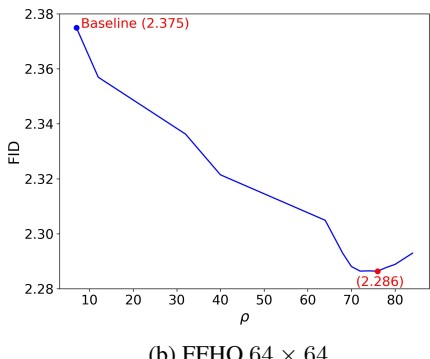
(b) FFHQ $64 \times 64$

Figure 8: Sampling schedule and sample quality. Compared to the baseline noise schedule proposed by EDM (i.e., $\rho = 7$), our method benefits from using a higher $\rho$ value. With a fixed number of sampling steps, this adjustment enables our sampling process to focus more on the later stages of the reverse process, where high-frequency detail are generated. As a result, the sample quality is improved across both CIFAR-10 and FFHQ datasets.

Table 7: Quantitative comparisons of FID and IS scores on unconditional CIFAR-10 for different BNR values, evaluated using the DDPM (Ho et al., 2020) architecture.

| Methods | BNR | FID ↓ | IS ↑ |
|---|---|---|---|
| DDPM (Ho et al., 2020) | 0 | 3.17 | 9.46 |
| Ours | 0.3 | 3.11 | 9.48 |
| Ours | 0.5 | **3.03** | **9.51** |

steps are allocated to generate high-frequency detail (i.e., the later stages of the reverse process), ultimately enhancing the quality of the generated samples.

## C.2 EXPERIMENTAL RESULTS ON DDPM ARCHITECTURE

In the main manuscript, we demonstrate the effectiveness of our proposed approach using the stronger baseline, improved DDPM++/NSCN++, introduced by Karras et al. (2022). In this section, we extend our experiments to the network architecture proposed in DDPM (Ho et al., 2020), The quantitative comparison on unconditional CIFAR-10 is presented in Tab. 7. Our method achieves a lower FID score and a higher IS, indicating improved image distribution modeling and enhanced sample quality. The consistent performance improvement across different network architectures further demonstrates the generalizability of our proposed method. Additionally, we conduct experiments with a lower BNR value, as reported in Tab. 7, and the trend of the results aligns with those presented in Tab. 3. These findings highlight the enhanced generation quality achieved by transitioning from Hot Diffusion to our proposed Warm Diffusion. This improvement stems from effectively leveraging the spectral dependency of images while addressing out-of-manifold issues by avoiding excessive blurring in the diffusion process.

## D ADDITIONAL RESULTS

In this section, we provide additional visual comparisons to present more qualitative results. Fig. 9 showcases generated samples for various BNR values, as discussed in Tab. 3. Fig. 10 illustrates samples produced by our re-implementation of Blurring Diffusion (Hoogeboom & Salimans, 2024), as outlined in Tab. 4. A visual comparison of different parameterization techniques, discussed in Tab. 5, is shown in Fig. 11. Furthermore, uncurated (non-cherry-picked) samples from all datasets used in our experiments are presented in Fig. 12.

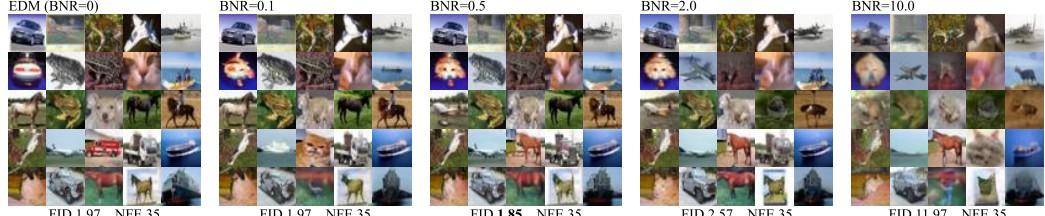

Figure 9: Visual comparison of different BNR values with a constant number of sampling steps. For cases where $BNR > 0.5$, using the same number of sampling steps, the generated samples tend to become blurrier as the BNR value increases, leading to poorer FID scores. This highlights the trade-off between blurring and noising, where overly prioritizing the blurring process can negatively impact sample quality.

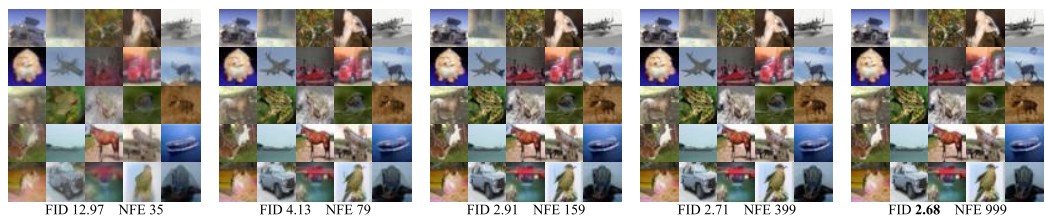

Figure 10: Visual results of our re-implemented Blurring Diffusion (Hoogeboom & Salimans, 2024) generated samples with different NFEs. As the number of sampling steps increases, the generated samples show progressively more detailed high-frequency components. Lower NFEs lead to blurrier samples due to insufficient reverse steps as discussed in Sec. 4.3 due to the data manifold shifts.

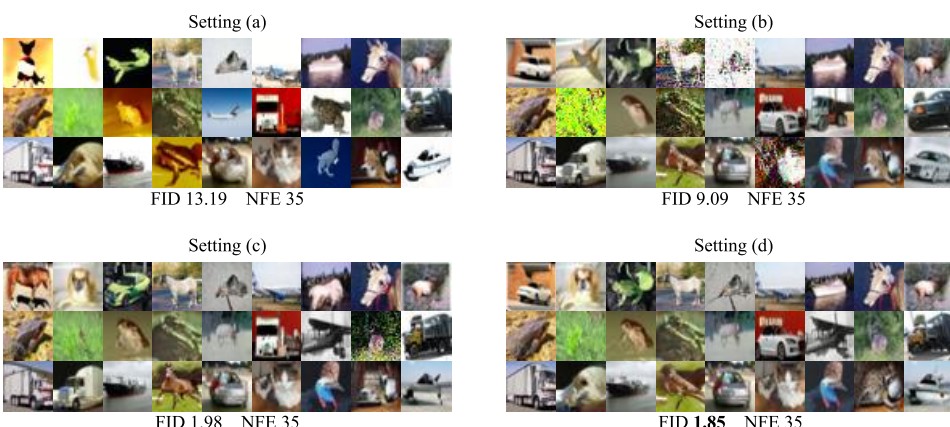

Figure 11: Visual comparison of different training objectives and parameterization techniques discussed in Tab. 5. Setting (a) and (b) either suffer from inaccurate predictions or amplified high-frequency signals, which lead to poorer generation quality. Although Setting (c) generates better results, it does not take advantage of the BNR adjustment to ease the neural network's learning process. In contrast, Setting (d), which employs a divide-and-conquer strategy, benefits from the BNR adjustment by better leveraging the spectral dependency of images. This leads to improved learning for the neural network and results in a noticeable improvement in sample quality.

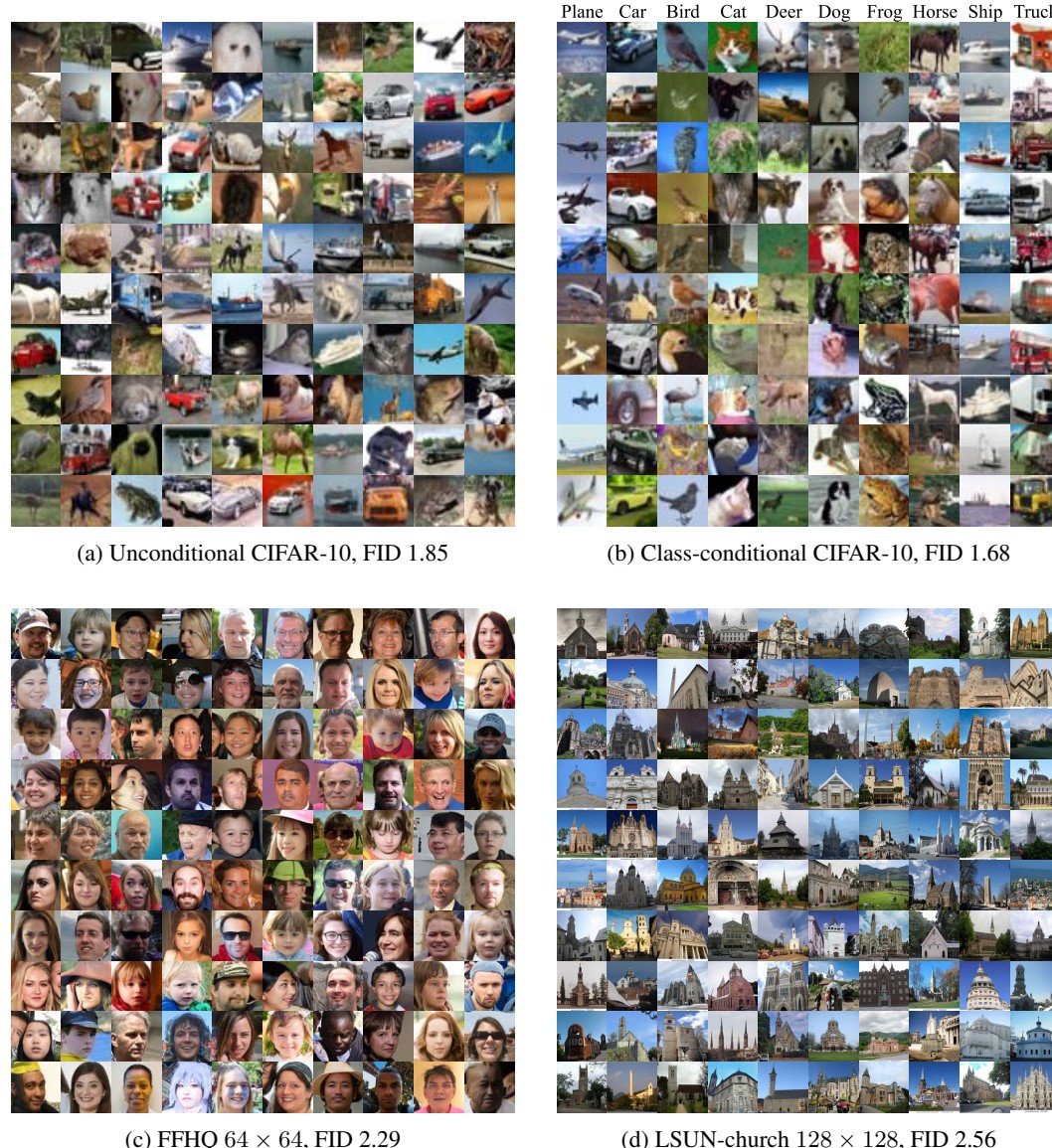

(a) Unconditional CIFAR-10, FID 1.85

(b) Class-conditional CIFAR-10, FID 1.68

(c) FFHQ $64 \times 64$, FID 2.29

(d) LSUN-church $128 \times 128$, FID 2.56

Figure 12: Uncurated samples from CIFAR-10, FFHQ $64 \times 64$, and LSUN-church $128 \times 128$.

