# OpenReview forum: "Warm Diffusion: Recipe for Blur-Noise Mixture Diffusion Models"
_ICLR.cc/2025/Conference — ICLR 2025 Poster_

### Official Review · Reviewer_W1SP · 2024-10-28

**Soundness:** 3
**Presentation:** 3
**Contribution:** 3
**Rating:** 6
**Confidence:** 5

**Summary:**

This paper proposes warm diffusion to bridge hot diffusion, which relies entirely on noise, and cold diffusion, which uses only blurring without noise. This paper claims that noise prediction has the property of good diversity and data manifold, and the blur prediction has the advantage of better spectral dependency and convergence. Then, the blur-to-noise ratio problem in proposed unified Blur-Noise Mixture Diffusion Model (BNMD) is discussed. The performance on the choosen datasets is promissing.

**Strengths:**

1. The concept of blur-noise mixture is interesting. The analyses of cold diffusion and hot diffusion are insightful. It is also suggested to verify the claims about these two diffusion paradigms with supporting evidence or visualization, which is missing in the current paper.
2. The analyses of the blur-to-noise ratio are clearly discussed, which is an important aspect in the proposed mixture paradigm.
3. The results on the conducted small-scale datasets are promissing.

**Weaknesses:**

1. This paper claims to decompose the training object into two parts: deblurring and denoising. However, the training target of the deblurring part is the residual. This is quite confusing, since this "deblurring" transition also involves denoising and is not even doing the deblurring job.
2. This two training target operation enforces the network to simultaneously modeling the distribution of the blur and the residual. Besides, the blur also has various levels. These may make the distribution modeling task harder than just predicting the noise/data, especially when the data scale is large.
3. The experimental results on large-scale datasets are missing, eg ImageNet.  Since the recent sota diffusion models usually conduct experiments on ImageNet and even larger-scale datasets, the results absence on these datasets make the actual performance and the superiority of this method ambiguous.

**Questions:**

My question mainly focuses on the training target and the corresponding optimization diffuculty. Refer to the weakness for details.
My final score depends on the response and I am willing to raise the score with convincing responses.

---

> ### Author Response · Authors · 2024-11-21
> **Q1. Regarding the "deblurring" component in our method**
>
> Thank you for the feedback regarding the "deblurring" component in our method. We understand your concern about the training target of the deblurring part and the potential confusion about its role in our workflow.
>
> Let me clarify by starting with the naive approach to solving the inverse problem in our setting. Suppose we have a blurry and noisy observation $x_t= blur(x) + noise$, and our goal is to recover the clean signal. A straightforward approach involves two stages:
> 1. **Denoising**: Use a neural network $D$ to remove noise from $x_t$ such that $D(x_t) \approx blur(x)$, recovering a blurry version of the clean signal.
> 2. **Deblurring**: Use another neural network $R$ to predict the missing high-frequency detail, $R(D(x_t)) \approx x - blur(x)$, from the denoised result $D(x_t)$. The residual is then added back to $D(x_t)$ to reconstruct the original clean image $x$.
>
> While this approach can be effective, it requires two separate networks and two sequential forward passes, making it computationally expensive and inefficient. To address this inefficiency, we propose a single unified framework that integrates these tasks. Instead of using two independent networks, we introduce a **shared backbone** $F$, which encodes the input observation $x_t$. This backbone supports two projection heads:
> * **Denoising head** $D$: Predict the blurry image, $blur(x)$, from the encoded features of $F$.
> * **Deblurring head** $R$: Predict the missing high-frequency residual, $x - blur(x)$, using the same encoded features from $F$.
>
> By leveraging this shared backbone and adopting a multi-task learning framework, our model effectively performs both denoising and deblurring in a **single forward pass**. The denoising head $D$ predicts the blurry image, while the deblurring head $R$ uses the same features—representing a blurry version of the image—to reconstruct the high-frequency details.
>
> This design achieves several advantages:
> 1. **Efficiency**: It reduces computational cost by requiring only a single network and a single forward pass, compared to the naive approach.
> 2. **Fairness**: It maintains comparable model complexity, ensuring a fair comparison with baselines.
> 3. **Effectiveness**: It allows the network to perform denoising and deblurring simultaneously, leveraging shared features to enhance overall performance.
>
> We hope this explanation clarifies the role of the "deblurring" task in our method and resolves the confusion regarding its interaction with denoising. Thank you again for raising this important point, which allowed us to provide a detailed clarification.

---

> ### Author Response · Authors · 2024-11-21
> **Q2. Regarding the potential "optimization difficulty" in our approach**
>
> Thank you for raising these insightful points. The decision to model both the blur and residual distributions in our approach stems from the goal of leveraging the spectral dependency inherent in images—where high-frequency details are closely correlated with low-frequency structures. By explicitly targeting these two components, our method enables the network to learn specialized functions for denoising and deblurring, enhancing both the robustness and interpretability of the model. This decomposition allows the network to better utilize low-frequency content as contextual guidance for reconstructing high-frequency details, ultimately improving the overall quality of the generated results.
>
> Moreover, this approach aligns with the concept of Single Input Multiple Outputs (SIMO), commonly used in multi-task learning frameworks (e.g., TCDCN [1]), which has been shown to improve both training efficiency and model accuracy. Instead of directly predicting a clean image, we decompose the task into predicting a blurry version of the input and the corresponding high-frequency residual. This decomposition introduces additional supervision signals through multi-task learning, enhancing explainability and increasing model robustness.
>
> On the other hand, we hope we have correctly understood the reviewer’s concern regarding the potential 'optimization difficulty' caused by the need to model various blur levels. If our understanding is accurate, the concern is that training a denoiser to produce different blurry outputs at various diffusion timesteps might be more challenging than training it to predict the noise/data (i.e., the clean target image), even though the noise levels vary across timesteps. To address this concern, we employ a similar strategy to standard diffusion models by including a time index as an input to the denoiser and specifying the corresponding target blur images as training targets. This enables the denoiser to learn distinct behaviors for each timestep effectively.
>
> Furthermore, as highlighted in the EDM [2] framework (see Fig. 1 in EDM), when a clean target image is used as the learning target to train the denoiser of a standard diffusion model, the well-trained denoiser’s output at different timesteps naturally resembles a blurry version of the target, with the blur level varying over time. Building on this finding, our approach directly assigns a corresponding blurry version of the clean image as the learning target for training the denoiser. This does not introduce additional difficulty, as the denoiser’s output inherently tends to be blurry. Consequently, this gradual adaptation—from heavily blurred to lightly blurred targets—makes our training objective potentially easier to optimize compared to conventional diffusion models, which use the clean target image as direct supervision at all timesteps.
>
> We hope this clarification addresses your concerns regarding the potential optimization difficulty in our approach. Thank you again for your valuable feedback, which has allowed us to better explain these key aspects of our method.
>
> Reference:
>
> [1] Zhanpeng Zhang, Ping Luo, Chen Change Loy, and Xiaoou Tang. Facial landmark detection by deep multi-task learning. In Computer Vision–ECCV 2014: 13th European Conference, Zurich, Switzerland, September 6-12, 2014, Proceedings, Part VI 13, pp. 94–108. Springer, 2014.
>
> [2] Tero Karras, Miika Aittala, Timo Aila, and Samuli Laine. Elucidating the design space of diffusion-based generative models, 2022. URL https://arxiv.org/abs/2206.00364.

---

> ### Author Response · Authors · 2024-11-21
> **Q3. Regarding the experimental results on "large-scale datasets"**
>
> Thank you for raising this important point regarding the absence of experimental results on large-scale datasets such as ImageNet. We understand that the lack of these results might raise questions about the scalability and superiority of our proposed method. Unfortunately, conducting experiments on large-scale datasets is currently beyond our computational resources. For instance, our baseline method, EDM [1], requires approximately 13 days of training on 32 NVIDIA A100 GPUs to train on the ImageNet 64×64 dataset. In contrast, our resources are limited to 8 NVIDIA RTX 4090 GPUs, making it impractical to replicate such large-scale experiments within our constraints. Additionally, the primary motivation of this work is to address a fundamental question: how to bridge the "hot diffusion" and "cold diffusion" paradigms while providing a deeper analysis of diffusion behaviors.
>
> Despite the aforementioned limitations, we have taken significant steps to validate the robustness and effectiveness of our method. Specifically, we have conducted experiments on multiple small-scale datasets with diverse characteristics, including variations in shared structures, resolution, and class labels. These datasets allow us to evaluate the generalizability and performance of our method under a range of conditions. The consistent and promising results across these datasets provide evidence for the effectiveness of our approach.
>
> While we acknowledge that results on large-scale datasets would further validate our method, we hope that our comprehensive experiments on smaller, diverse datasets effectively demonstrate its potential and reinforce our insights into bridging the "hot diffusion" and "cold diffusion" paradigms. Thank you again for your understanding and for raising this point, which highlights an important avenue for future exploration.
>
> Reference:
>
> [1] Tero Karras, Miika Aittala, Timo Aila, and Samuli Laine. Elucidating the design space of diffusion-based generative models, 2022. URL https://arxiv.org/abs/2206.00364.

---

### Official Review · Reviewer_GSiJ · 2024-10-31

**Soundness:** 3
**Presentation:** 2
**Contribution:** 3
**Rating:** 6
**Confidence:** 3

**Summary:**

The authors analyze the characteristics and limitations of hot diffusion (noise-based) and cold diffusion (blur-based) and propose a unified framework that combines both diffusion processes. This approach effectively mitigates the shortcomings of each model, achieving complementary advantages. The authors also conduct extensive experiments and data analyses to demonstrate the superiority of the proposed method.

**Strengths:**

1. The authors analyze the characteristics and limitations of both hot and cold diffusion, leading to a unified diffusion architecture that combines both approaches.
2. The authors introduce the new concept of Blur-to-Noise Ratio (BNR), which enables better analysis of diffusion models.
3. Extensive quantitative and qualitative analyses, including comparisons with state-of-the-art methods and detailed data analysis, are provided.
4. Experimental results demonstrate the effectiveness of the proposed method in image generation.

**Weaknesses:**

1. In Figure 1, the authors do not sufficiently explain the meaning of each module. For example, the significance of different-sized circles in the left chart is unclear, as well as the meaning of “Data manifold (indexed by noise level)” and the images depicted. It is suggested that the caption be revised to simplify understanding.
2. The authors use the improved DDPM++/NCSN++. However, it would be beneficial to experiment with the proposed approach on other baseline architectures, such as the original DDPM [1]. This would help demonstrate the generalizability of the method beyond specific model improvements.

[1] Ho J, Jain A, Abbeel P. Denoising diffusion probabilistic models. In NeurIPS, 2020.

**Questions:**

1. Please provide a more detailed explanation of the images in Figure 1.
2. Provide experimental results using DDPM as a baseline, showing the performance improvements of the proposed method compared to the original model and alternative approaches.

---

> ### Author Response · Authors · 2024-11-25
> **Q1. Please provide a more detailed explanation of the images in Figure 1**
>
> Thank you for pointing this out, and we sincerely apologize for any confusion caused by the figure. We have addressed the concerns and made improvements in the revised manuscript.
>
> 1. **Circle Sizes in Figure 1**
>
>     In the original Figure 1, each circle was intended to represent a specific state in the two-pronged diffusion process. The variation in circle sizes was unintentional and did not carry any specific meaning. To avoid potential misunderstandings, we have corrected this in the revised figure by ensuring that all circles are uniform in size.
>
> 2. **Explanation of "Data Manifold (Indexed by Noise Level)" and Depicted Images**
>
>     The section of Figure 1 labeled "data manifold shift" visualizes the data manifolds of two diffusion processes under different blur-to-noise ratio (BNR) settings: lower BNR (left) and higher BNR (right). This visualization highlights how the BNR setting influences the shape of the data manifold, leading to a manifold shift. One of our key research motivations is to understand how these manifold changes affect the diffusion process, as discussed in Section 4.3 and illustrated in Figure 4.
>
>     - **High BNR** vs. **Low BNR**:
>
>         Under high BNR, the data manifold shift leads to out-of-manifold issues during diffusion sampling. In contrast, low BNR settings exploit the correlation between high-frequency image details and low-frequency structures (spectral dependency) less effectively. This trade-off between high and low BNR is one of the motivations for our study.
>
> 3. **Definition of Data Manifold**
>
>     The "data manifold" in this context refers to the probability distribution of the two-pronged diffusion process over diffusion timesteps. As the forward process progresses, increasing levels of noise and blur are applied to the data, causing the distribution to transition from the original data distribution to a Gaussian distribution. This gradual transition represents how the data becomes increasingly blurred and noisy over time.
>
>     - **Impact of High BNR**:
>
>         Under higher BNR, distinct high-frequency details are progressively filtered out due to the blurring operation, leaving only shared low-frequency content. This leads to a faster convergence into a single-mode Gaussian distribution (as illustrated by the merging of red and blue curves in the section of Figure 1 labeled "data manifold shift").
>
>         -  The red and blue lines in the figure represent the means of Gaussian distributions derived from samples with shared low-frequency structures but differing high-frequency details over the diffusion timesteps. As the diffusion process progresses, the high-frequency differences are gradually filtered out, leaving only the shared low-frequency characteristics. This convergence visually demonstrates the effect of the diffusion process.
>
> 4. **Revisions to Figure 1 and Caption**
>
>     To address these points, we have improved both the figure and its caption in the revised manuscript to make these explanations clearer and to ensure the figure accurately represents the concepts discussed.
>
> We hope these clarifications make Figure 1 more comprehensible and highlight the research motivations underlying our work. Thank you for your feedback, which has helped us improve the presentation of this important aspect of our study.

---

> ### Author Response · Authors · 2024-11-25
> **Q2. Regarding experimental results using DDPM as a baseline**
>
> Thank you for your valuable comment and for raising the concern regarding the generalizability of our proposed method across different network architectures. To address this, we conducted additional experiments by applying our proposed approach to the baseline DDPM [1] architecture. These experiments were evaluated under the unconditional CIFAR-10 setting, and the results are presented below:
>
> | ** Methods**   | **BNR**         | **FID $\downarrow$** | **IS $\uparrow$**  |
> |---------------------|--------------------|---------|---------|
> | DDPM [1]           | 0  | 3.17    | 9.46    |
> | Ours               | 0.3    | 3.11    | 9.48    |
> | Ours               | 0.5   | **3.03**    | **9.51**    |
>
> As shown in the table, our method demonstrates **improved performance** in both FID and IS metrics under the original DDPM [1] architecture. These results highlight the effectiveness of our approach in accurately modeling the image distribution and generating higher-quality samples, underscoring its generalizability beyond specific model improvements. Furthermore, the results are consistent with our observations when applying the method to both DDPM and the improved DDPM++. Specifically, generation quality consistently improves as higher BNR values are applied, **provided the BNR remains below 0.5**. This indicates that our method successfully balances leveraging the spectral dependency of images with mitigating the out-of-manifold issue that can arise when incorporating blurring into the diffusion process.
>
> Additionally, researchers [2,3,4] have made significant advancements **in improving the architecture of diffusion models**, resulting in several variations over recent years. However, most of these efforts focus on the framework of the hot diffusion process. A central claim of our paper is that the hot diffusion process may have inherent limitations in modeling image distributions, as it overlooks the spectral dependency present in images. To address this gap, we introduce a blur-noise mixture diffusion process that better captures the correlation between high- and low-frequency signals in images. Moreover, our approach resolves the out-of-manifold issues associated with incorporating blurring, **offering a complementary perspective to existing work focused on architectural enhancements**.
>
> We hope this explanation addresses concerns regarding the generalizability of our method. The experimental results have been included in the supplementary material of the revised manuscript for further reference. Thank you again for your insightful feedback, which has greatly contributed to the completeness of our work.
>
> Reference:
>
> [1] Jonathan Ho, Ajay Jain, and Pieter Abbeel. Denoising diffusion probabilistic models, 2020. URL https://arxiv.org/abs/2006.11239.
>
> [2] Alex Nichol and Prafulla Dhariwal. Improved denoising diffusion probabilistic models, 2021. URL https://arxiv.org/abs/2102.09672.
>
> [3] Yang Song, Jascha Sohl-Dickstein, Diederik P. Kingma, Abhishek Kumar, Stefano Ermon, and Ben Poole. Score-based generative modeling through stochastic differential equations, 2021. URL https://arxiv.org/abs/2011.13456.
>
> [4] Tero Karras, Miika Aittala, Timo Aila, and Samuli Laine. Elucidating the design space of diffusion-based generative models, 2022. URL https://arxiv.org/abs/2206.00364.

---

### Official Review · Reviewer_tjvC · 2024-11-04

**Soundness:** 3
**Presentation:** 3
**Contribution:** 2
**Rating:** 6
**Confidence:** 3

**Summary:**

The paper proposes a new diffusion framework named "Warm Diffusion," which introduces a unified Blur-Noise Mixture Diffusion Model (BNMD) to bridge the gap between hot diffusion (using noise) and cold diffusion (using blurring) models. The authors argue that both hot and cold diffusion paradigms have inherent limitations: hot diffusion ignores the correlation between image structures, while cold diffusion neglects the importance of noise in modeling the data manifold. Warm Diffusion leverages a Blur-to-Noise Ratio (BNR) to integrate both noise and blurring in a "divide-and-conquer" approach, enhancing model learning and image quality. Extensive experiments show that the method outperforms existing diffusion-based generative models on several benchmarks.

**Strengths:**

1. The idea of combining Blur and Noise is impressive as it exploits spectral dependencies of images while preserving the data manifold.

2. The proposed BNMD framework is evaluated across several benchmarks, including CIFAR-10, FFHQ, and LSUN-church datasets, sufficiently verifying the effectiveness of the proposed method.

**Weaknesses:**

1. I think the motivation of the work is not very clearly presented in the introduction section. The exact necessity of introducing blurring into the diffusion model should be given, which is the most important motivation of this work. Figure 5 shows that a lower BNR brings better FID results, which somehow seems to say that “blur” does not help to improve the quality of the generated image.

2. For the sampling process presented in algorithm 2, the sampling starts from a zero-mean Gaussian distribution. How can this be revealed from Eq. 4 (at time T)?

3. As shown in Table 5, there exist large deviations among the results produced by different parameterizations. More explanation need to be provided.

**Questions:**

Q1. Although the authors repeatedly mention “spectral dependency”, the definition of this term is not clear. I can't clearly understand what exactly the “strong correlation” between high-frequency and low-frequency structures is. In Figure 3, why “shifting more responsibility to the deblurring task” means “effectively utilizing the spectral dependency of images”?

Q2. In regard to blur-to-noise-ratio (BNR) defined by this work, it seems that the noise level and the blur level do not change for all time t. I want to further confirm that mentioning the effects of BNR, does the proposed diffusion model actually have a sequence of (a_1,…,a_T) and (b_1,…,b_T) or just one a and b for all timesteps?

---

> ### Author Response · Authors · 2024-11-25
> **Q1. Regarding the definition of "spectral dependency"**
>
> Thank you for highlighting the need for a clearer definition of "spectral dependency." To elaborate, this term refers to the relationship between the high-frequency and low-frequency components of an image. For example, consider a blurry image that lacks high-frequency details. Despite this, one can often infer the missing details based on the low-frequency content. If the blurry image depicts a repetitive structure, such as a brick wall, the low-frequency patterns provide strong contextual clues that allow us to reasonably deduce the finer textures of the bricks (i.e., the missing high-frequency details). This interconnected relationship—where high-frequency details are closely correlated with low-frequency structures—is what we define as "spectral dependency" in our paper.
>
> Regarding Figure 3, when we state "shifting more responsibility to the deblurring task," we refer to our divide-and-conquer strategy. Instead of treating the neural network as a black-box function that directly predicts a clean image, we decompose the problem into two tasks: denoising and deblurring. The **denoising task** focuses on recovering a blurry image obscured by noise (not directly reconstructing the clean target), while the **deblurring task** specifically predicts the missing high-frequency details by leveraging the information embedded in the low-frequency structures. This design explicitly encourages the network to learn and exploit the connection between high- and low-frequency components—i.e., the spectral dependency of images. Next, by assigning suitable responsibility to the deblurring task, the network more effectively learns the correlation between high- and low-frequency signals, thereby better utilizing the spectral dependency of images. At the same time, this approach reduces the burden of the standard one-step design, which requires a denoising network to remove noise and recover the clean image simultaneously. Moreover, this divide-and-conquer framework enhances the interpretability of the network and improves its capacity to reconstruct images by explicitly leveraging this inherent property of image data.
>
> We hope this clarification addresses the concerns regarding the definition of "spectral dependency" and the connection between deblurring and utilizing spectral dependency.

---

> ### Author Response · Authors · 2024-11-25
> **Q2. Regarding the effects of BNR and the proposed diffusion process**
>
> Thank you for raising this question and giving us the opportunity to clarify the concept of the blur-to-noise ratio (BNR) and the design of our diffusion process.
>
> In our proposed approach, we **indeed define a sequence of parameters**, $(\alpha_1, \dots, \alpha_T)$ and $(\beta_1, \dots, \beta_T)$, that vary over time, as detailed in Section 4.1. These sequences are **monotonically increasing**, controlling the levels of blur and noise at each timestep. This dynamic adjustment ensures a gradual and systematic progression of both components throughout the diffusion process. By the final timestep $T$, the noise level becomes sufficiently high $(\beta_T \gg \beta_{data})$, rendering the distribution nearly indistinguishable from a zero-mean Gaussian, aligning with observations in prior works such as SMLD [1], Score-SDE [2], and EDM [3].
>
> While $\alpha_t$ and $\beta_t$ evolve across timesteps, the blur-to-noise ratio (BNR) remains **“Constant”** throughout the process. This means that although the **absolute levels of blur and noise change dynamically** according to $\alpha_t$ and $\beta_t$, their **ratio is fixed** at each timestep. As shown in Figure 1(a), varying the BNR (a tunable constant) enables us to explore different blur-noise mixture diffusion processes.
>
>
> For example, when BNR is set to 0, the sequence $(\alpha_1, \dots, \alpha_T)$ remains zero throughout, while $(\beta_1, \dots, \beta_T)$ forms an increasing sequence. This corresponds to a standard denoise-based diffusion process, known as "Hot Diffusion," where no blurring is applied. As the BNR increases, higher levels of blurring are applied relative to the same noise level, transitioning the diffusion process from (1) Hot Diffusion to (4) Cold Diffusion, as depicted by the “red → yellow → green → blue” sequences in Figure 1(a).
>
> We hope this explanation clarifies the definition of BNR and the mechanics of our proposed diffusion process.
>
> Reference:
>
> [1] Yang Song and Stefano Ermon. Generative modeling by estimating gradients of the data distribution. In H. Wallach, H. Larochelle, A. Beygelzimer, F. d'Alche-Buc, E. Fox, and ´ R. Garnett (eds.), Advances in Neural Information Processing Systems, volume 32. Curran Associates, Inc., 2019. URL https://proceedings.neurips.cc/paper_files/paper/2019/file/3001ef257407d5a371a96dcd947c7d93-Paper.pdf.
>
> [2] Yang Song, Jascha Sohl-Dickstein, Diederik P. Kingma, Abhishek Kumar, Stefano Ermon, and Ben Poole. Score-based generative modeling through stochastic differential equations, 2021. URL https://arxiv.org/abs/2011.13456.
>
> [3] Tero Karras, Miika Aittala, Timo Aila, and Samuli Laine. Elucidating the design space of diffusion-based generative models, 2022. URL https://arxiv.org/abs/2206.00364.

---

> ### Author Response · Authors · 2024-11-25
> **W1. Regarding our "motivation" and the meaning of "Figure 5"**
>
> Thank you for raising this concern. We appreciate the opportunity to elaborate on the motivation behind introducing blurring into the diffusion model and to clarify the interpretation of Figure 5.
>
> The primary motivation for incorporating blur lies in the **spectral dependency** of images—the strong correlation between low-frequency structures (e.g., general shapes) and high-frequency detail (e.g., fine textures). Traditional "hot" diffusion processes treat all frequency components equally, thereby overlooking this intrinsic property of images. By explicitly introducing blur into the diffusion process, our method leverages this spectral dependency, enabling the network to first denoise to a blurry image and then deblur to recover the missing detail. This two-step approach allows the model to reconstruct high-frequency detail more effectively, using low-frequency content as a contextual guide.
>
> Regarding Figure 5, it should not be interpreted as evidence that “blur does not improve quality.” Instead, the figure highlights the limitations of selecting an excessively high blur-to-noise ratio (BNR) and illustrates how performance declines due to the out-of-manifold issue, as discussed in Section 4.3. Our spectral analysis (Section 4.4) reveals that when the BNR exceeds 0.5, the data manifold begins to shift, leading to out-of-manifold problems when the same number of sampling steps is used (see Figure 4). To explore this, Figure 5 investigates cases with BNR ≥ 0.5, examining whether increasing the number of sampling steps can mitigate these challenges. The results confirm that higher BNR settings indeed require more sampling steps to achieve optimal performance, aligning with our analysis.
>
> In contrast, Table 3 demonstrates that when the BNR is below 0.5, sample quality consistently improves as the BNR increases. This improvement occurs because the data manifold is preserved, allowing the model to better exploit the spectral dependency of images. Specifically, the network learns the correlation between high- and low-frequency signals more effectively. These findings underscore that blur, when appropriately balanced with noise, enhances sample quality by aligning the diffusion process with the intrinsic properties of image data.
>
> We hope this explanation clarifies both the motivation for introducing blur and the interpretation of Figure 5. Thank you again for giving us the opportunity to address this important point.

---

> ### Author Response · Authors · 2024-11-25
> **W2. Regarding the "sampling process" of our approach**
>
> Our proposed diffusion process is governed by two sequences of parameters, $(\alpha_1,\dots,\alpha_T)$ and $(\beta_1,\dots,\beta_T)$, which monotonically increase over time to control the levels of blur and noise, respectively, as detailed in Section 4.1. In Eq. (4), as time progresses, the distribution becomes increasingly influenced by high levels of blur and noise. By the final timestep $T$, the noise level becomes sufficiently high $(\beta_T \gg \beta_{data})$. For instance, following EDM [1], we set $\beta_T = 80$ while normalizing the image data range to $\[-1,1\]$. Under these settings, the distribution at $T$ becomes nearly indistinguishable from a zero-mean Gaussian, consistent with observations in prior works such as SMLD [1], Score-SDE [2], and EDM [3].
>
> As a result, the sampling process in Algorithm 2 starts from a zero-mean Gaussian distribution because it mirrors the distribution reached at the end of the forward process. During the reverse process, the sequences $(\alpha_T,\alpha_{T-1},\dots,\alpha_0)$ and $(\beta_T, \beta_{T-1},\dots,\beta_0)$ are used to iteratively perform deblurring and denoising. At each step, the noise in the current observation is reduced while more high-frequency details are introduced by lowering the blur level. This process gradually transforms the Gaussian distribution back into the original data distribution.
>
> We hope this explanation clarifies why the sampling process begins with a zero-mean Gaussian distribution and how the sampling process functions.
>
> Reference:
>
> 1] Yang Song and Stefano Ermon. Generative modeling by estimating gradients of the data distribution. In H. Wallach, H. Larochelle, A. Beygelzimer, F. d'Alche-Buc, E. Fox, and ´ R. Garnett (eds.), Advances in Neural Information Processing Systems, volume 32. Curran Associates, Inc., 2019. URL https://proceedings.neurips.cc/paper_files/ paper/2019/file/3001ef257407d5a371a96dcd947c7d93-Paper.pdf.
>
> [2] Yang Song, Jascha Sohl-Dickstein, Diederik P. Kingma, Abhishek Kumar, Stefano Ermon, and Ben Poole. Score-based generative modeling through stochastic differential equations, 2021. URL https://arxiv.org/abs/2011.13456.
>
> [3] Tero Karras, Miika Aittala, Timo Aila, and Samuli Laine. Elucidating the design space of diffusion-based generative models, 2022. URL https://arxiv.org/abs/2206.00364.

---

> ### Author Response · Authors · 2024-11-25
> **W3. Regarding experimental results of Table 5**
>
> Thank you for raising this point regarding the large deviations observed in Table 5.
>
> The purpose of Table 5 is to provide a comprehensive comparison of different parameterization strategies to better illustrate the advantages of our divide-and-conquer approach. Each parameterization introduces specific trade-offs, and the observed deviations reflect how effectively the design addresses the denoising and deblurring processes separately. To complement the quantitative data, we provide visual examples in the supplementary material (Figure 11) to clarify the observed differences. Below, we elaborate on the results:
>
> 1. **Table 5(a): Single-branch approach predicting the clean signal**
>
>     Using a single branch to directly predict the clean signal performs poorly because the network is tasked with simultaneously handling both denoising and deblurring, leading to high optimization complexity. This often results in unnatural artifacts, such as strange color patterns, as shown in Figure 11(a). These artifacts highlight the challenges of addressing both tasks in a single branch.
>
> 2. **Table 5(b): Predicting a blurry image as the learning target**
>
>     Changing the learning target to a blurry image yields slightly better performance. Predicting a blurry image is inherently easier, as it requires less fine-grained detail compared to predicting a clean image. However, this approach introduces a major drawback: the predicted blurry signal lacks accurate high-frequency detail. When the deblurring step attempts to recover the missing high-frequency content, any inaccuracies in the prediction are amplified, leading to noisy patterns in the generated samples, as illustrated in Figure 11(b).
>
> 3. **Table 5(c): Two-branch design**
>
>     Employing a two-branch design improves upon the previous approaches. Here, one branch focuses on uncovering low-frequency content (denoising), while the other attempts to directly predict the clean signal (denoising and deblurring). This separation reduces optimization complexity, resulting in more realistic colors and fewer noisy artifacts, as demonstrated in Figure 11(c). However, the performance is only comparable to the baseline diffusion model (e.g., EDM). This approach does not fully leverage the spectral dependency of images, as it lacks a properly designed learning target that aligns with the inherent correlation between low- and high-frequency components.
>
> 4. **Proposed Method (Table 5(d)): Divide-and-conquer strategy**
>
>     Our proposed divide-and-conquer strategy explicitly forces the network to model the spectral dependency of images. By decoupling the tasks of denoising (recovering low-frequency content) and deblurring (predicting high-frequency residuals) and aligning these tasks with well-structured learning objectives, the model achieves significant improvements in sample quality. This is evident in both the quantitative results in Table 5 and the high-quality visuals shown in Figure 11(d).
>
> In summary, the deviations observed across parameterizations in Table 5 stem from differences in how effectively the neural network utilizes the intrinsic properties of images and the potential numerical challenges in reconstructing a less blurry image from a blurry one. Our proposed method excels by explicitly leveraging the spectral dependency of images and designing the learning process to align with this property. We hope this explanation, along with the referenced visual examples, addresses your concerns.

---

> ### Comment · Reviewer_tjvC · 2024-11-26
> **Response to authours.**
>
> Thanks for your detailed response, which has addressed all my concerns well, particularly on W3. Thus, I have increased my score to 6.

---

> > ### Author Response · Authors · 2024-11-29
> > **Response to Reviewer tjvC**
> >
> > Thank you for your positive feedback and for carefully reviewing our work. We greatly appreciate your thoughtful comments, which have helped improve our draft, and are pleased that our explanations addressed your concerns. Your increased score is highly encouraging, and we sincerely thank you for your support.

---

### Official Review · Reviewer_iqF4 · 2024-11-04

**Soundness:** 2
**Presentation:** 2
**Contribution:** 2
**Rating:** 5
**Confidence:** 3

**Summary:**

This paper proposes Warm Diffusion, a unified Blur-Noise Mixture Diffusion Model (BNMD), to control blurring and noise jointly.

**Strengths:**

The paper is well-written

**Weaknesses:**

1.An improvement of 1-2 points in FID does not result in any noticeable change in visual effects. In fact, the actual visual quality may not be better than those with lower FID scores.

2.I hope the authors can present results that are sufficiently stunning or impactful. There are currently many papers in this area, and everyone is focused on slightly improving FID and IS, but the visual quality is still much worse than the current FLUX. This leaves me with no motivation to decide whether to accept any of these papers.

**Questions:**

see above

---

> ### Author Response · Authors · 2024-11-25
>
> Thank you for your valuable feedback. We recognize that many large-scale foundation models have been proposed and have achieved impressive generation quality compared to existing frameworks. The success of these models can be attributed to several factors, such as the use of large-scale datasets, complex and expansive model architectures, and various training tricks —all within the framework of the hot diffusion process. However, training such large-scale foundation models demands immense computational resources. For instance, training SD3 (Stable Diffusion 3) [1] reportedly costs approximately $10 million, which is far beyond the reach of most researchers.
>
> Given the context, our work prioritizes a fundamental investigation of the design of diffusion models rather than directly competing with large-scale models in terms of visual quality. Specifically, while most researchers focus on the "hot diffusion" process—using denoising to model data distributions—few have explored incorporating general corruptions, such as blurring, into the diffusion process. Existing methods [2,3,4] that employ blurring often struggle to generate high-quality images, creating a notable performance gap compared to the standard noise-based diffusion process.
>
> The deblurring operation inherently depends on the spectral dependency of images (i.e., the relationship between high- and low-frequency components) during the generation process. However, the reasons behind the performance drop observed when integrating blurring into the diffusion process remain underexplored. Our work addresses this gap by:
>
> 1. Investigating the interplay between blur and noise during the generation process.
> 2. Analyzing the root causes of performance drops in previous blurring-based approaches.
> 3. Proposing a hybrid diffusion strategy that balances the benefits of deblurring and denoising.
>
> Through this exploration, we aim to strike a balance between Hot and Cold Diffusion [2], achieving improved image distribution modeling. Our experimental results across diverse datasets—featuring variations in structure, resolution, and class labels—consistently demonstrate significant improvements in FID and IS scores. These results provide strong evidence of the effectiveness of our proposed approach.
>
> Reference:
>
> [1] Patrick Esser, Sumith Kulal, Andreas Blattmann, Rahim Entezari, Jonas M¨uller, Harry Saini, Yam Levi, Dominik Lorenz, Axel Sauer, Frederic Boesel, Dustin Podell, Tim Dockhorn, Zion En- glish, Kyle Lacey, Alex Goodwin, Yannik Marek, and Robin Rombach. Scaling rectified flow transformers for high-resolution image synthesis, 2024. URL https://arxiv.org/abs/2403.03206.
>
> [2] Arpit Bansal, Eitan Borgnia, Hong-Min Chu, Jie S. Li, Hamid Kazemi, Furong Huang, Micah Gold-blum, Jonas Geiping, and Tom Goldstein. Cold diffusion: Inverting arbitrary image transforms without noise, 2022. URL https://arxiv.org/abs/2208.09392.
>
> [3] Severi Rissanen, Markus Heinonen, and Arno Solin. Generative modeling with inverse heat dissipation, 2023. URL https://arxiv.org/abs/2206.13397.
>
> [4] Emiel Hoogeboom and Tim Salimans. Blurring diffusion models, 2024. URL https://arxiv.org/abs/2209.05557.

---

### Meta-Review · Area_Chair_8um5 · 2024-12-17

**Metareview:**

This paper proposes a new diffusion model called Warm Diffusion that combines noise and blur degradation processes. The authors argue that existing diffusion models, which rely solely on noise or blur, have limitations. Noise-based models (hot diffusion) fail to exploit the correlation between high-frequency and low-frequency information in images, while blur-based models (cold diffusion) neglect the role of noise in shaping the data manifold.

Warm Diffusion uses a Blur-to-Noise Ratio (BNR) to control the balance between noise and blur. This approach simplifies score model estimation by disentangling the denoising and deblurring processes. The authors also analyze the BNR using spectral analysis to investigate the trade-off between model learning dynamics and changes in the data manifold.

Experiments on image generation benchmarks show the effectiveness of the proposed approach. The authors provide a comprehensive comparison of different parameterization strategies to illustrate the advantages of their divide-and-conquer strategy. They also conduct additional experiments to address concerns about the generalizability of their method across different network architectures.

Strengths:
- The paper is well-written and presents a novel approach to diffusion modeling.
- The proposed method addresses the limitations of existing diffusion models by combining noise and blur degradation processes.
- The authors provide a comprehensive analysis of the BNR and its impact on model performance.
- The experimental results support the effectiveness of the proposed approach.
- The authors have been responsive to the reviewers' concerns and have provided satisfactory clarifications and additional experiments.

Weaknesses:
- Limited novelty: There is already some work addressing image generation by combining noise and blur degradations.
- The experimental results are limited to small-scale datasets due to computational constraints.
- The paper could benefit from additional visualizations to support the claims about the properties of hot and cold diffusion.
- Some reviewers found the explanation of the deblurring component to be confusing.

Despite the limitations mentioned above, the paper presents a novel and promising approach to diffusion modeling. The proposed method addresses the limitations of existing models and has been shown to be effective on several benchmarks. The authors have also been responsive to the reviewers' concerns and have provided satisfactory clarifications and additional experiments. Overall, the paper makes a valuable contribution to the field.

**Additional Comments On Reviewer Discussion:**

During the rebuttal period, the reviewers raised several concerns, including the clarity of the motivation, the definition of spectral dependency, the interpretation of Figure 5, the sampling process, the large deviations in Table 5, the absence of experimental results on large-scale datasets, and the generalizability of the method across different network architectures.
The authors responded to each of these concerns in a comprehensive and satisfactory manner. They provided additional explanations, clarifications, and experimental results to address the reviewers' concerns.

For the final decision, I weighed each point raised by the reviewers and the authors' responses. I found that the authors have been responsive to the reviewers' concerns and have provided satisfactory clarifications and additional experiments. Overall, despite being on a borderline rating, I believe the paper makes a valuable contribution to the field and is deserving of acceptance.

---

### Decision · Program_Chairs · 2025-01-22

Accept (Poster)